# Type-2 innate lymphoid cells control the development of atherosclerosis in mice

Stephen A. Newland[1], Sarajo Mohanta[2], Marc Clément[1], Soraya Taleb[3], Jennifer A. Walker[4], Meritxell Nus[1], Andrew P. Sage[1], Changjun Yin[2], Desheng Hu[5], Lauren L. Kitt[1], Alison J. Finigan[1], Hans-Reimer Rodewald[6], Christoph J. Binder[7], Andrew N.J. McKenzie[4], Andreas J. Habenicht[2] & Ziad Mallat[1,3]

Type-2 innate lymphoid cells (ILC2) are a prominent source of type II cytokines and are found constitutively at mucosal surfaces and in visceral adipose tissue. Despite their role in limiting obesity, how ILC2s respond to high fat feeding is poorly understood, and their direct influence on the development of atherosclerosis has not been explored. Here, we show that ILC2 are present in para-aortic adipose tissue and lymph nodes and display an inflammatory-like phenotype atypical of adipose resident ILC2. High fat feeding alters both the number of ILC2 and their type II cytokine production. Selective genetic ablation of ILC2 in $Ldlr^{-/-}$ mice accelerates the development of atherosclerosis, which is prevented by reconstitution with wild type but not $Il5^{-/-}$ or $Il13^{-/-}$ ILC2. We conclude that ILC2 represent a major innate cell source of IL-5 and IL-13 required for mounting atheroprotective immunity, which can be altered by high fat diet.

[1] Department of Medicine, Division of Cardiovascular Medicine, University of Cambridge, Cambridge CB2 0SZ, UK. [2] Institute for Cardiovascular Prevention, Ludwig-Maximilians-University (LMU), 80336 Munich, Germany. [3] Institut National de la Santé et de la Recherche Médicale, U970 Paris, France. [4] Division of Protein and Nucleic Acid Chemistry, MRC Laboratory of Molecular Biology, Cambridge CB2 0QH, UK. [5] State Key Laboratory of Molecular Vaccinology and Molecular Diagnostics, Xiamen University, Xiamen, Fujian 361102, China. [6] Division of Cellular Immunology, German Cancer Research Center, 69120 Heidelberg, Germany. [7] Department of Laboratory Medicine, Medical University of Vienna and Center for Molecular Medicine (CeMM) of the Austrian Academy of Sciences, 1090 Vienna, Austria. Correspondence and requests for materials should be addressed to Z.M. (email: zm255@medschl.cam.ac.uk).

Cardiovascular disease is the leading cause of death worldwide, increasing in incidence year on year and was accountable for one in four deaths globally in 2010 (ref. 1). Atherosclerosis is the major cause of cardiovascular disease where deposits of low-density lipoproteins in the arterial wall lead to the infiltration of immune cells, inflammation and growth of fibro-fatty plaques. This process can culminate in occlusion of the artery following plaque disruption and thrombosis[2].

Plaque maturation is influenced by the populations of innate and adaptive immune cells infiltrating the lesion, their activation state and how they communicate with non-immune cells in the surrounding arterial tissue[3–5]. Hypercholesterolaemia and high fat diet (HFD) also trigger systemic immune responses that modulate the atherosclerotic process, which may explain the profound impact of spleen-dependent responses on several aspects of the atherosclerotic immune response[6–8].

Innate lymphoid cells (ILC) are a rare cell population that are closely related to T and B lymphocytes, but which do not express recombined antigen receptors such as the T-cell receptor and B-cell receptor. Early research identified many different subtypes including conventional natural killer (NK) cells[9], lymphoid tissue inducer cells[10,11], nuocytes[12] and natural helper cells[13]. ILC can be assigned to one of three groups, ILC1, ILC2 or ILC3 (ref. 14). These mirror the T helper (Th)1, Th2 and Th17 paradigm of T-cell biology and share effector cytokines and transcription factors. Th1 cells promote atherogenesis[4], which is also the case for ILC1-related NK cells[15]. However, the impact of Th2 and Th17 bias on the atherosclerotic process is more complex; they may either enhance or limit the disease[4,16].

ILC2 were initially identified as an innate source of IL-13 during helminth infection[12]. Subsequently they have been observed secreting large quantities of type II cytokines (IL-5, IL-13, IL-9), regulating innate and adaptive immune responses in several inflammatory settings (reviewed in ref. 17), modulating wound healing/tissue repair[18], and influencing adipose tissue function and metabolic homeostasis[19]. Furthermore, there is growing evidence that some type II cytokines are protective in mouse models of atherosclerosis. For example, IL-13 has been shown to protect from lesion development and promote plaque stability by increasing collagen deposition, and skewing the macrophage infiltrate towards an alternative activated phenotype[20]. IL-5 on the other hand may be protective via increasing titres of natural IgM antibodies specific for modified LDL epitopes[21]. Finally, the atheroprotective cytokines IL-33 (ref. 22) and IL-25 (ref. 23) can drive expansion of ILC2 (refs 24,25) and these cells may provide a crucial component of the protective mechanism. However, IL33 and IL-25 activate may other cellular responses independently of ILC2, and type II cytokines are also secreted by other cell types and may act on atherosclerosis independently of ILC2.

Two recent studies suggested that ILC2 expansion in mice may have an athero-protective role[23,26]. However, the results were based on pharmacologic expansion of an ILC2 population, sometimes in immunodeficient mice, and were confounded by dramatic alterations in plasma cholesterol levels after treatment, or by alterations in other immune cell populations. Another study showed that total deficiency of Id3, which leads to increased atherosclerosis in mice, may reduce IL-5 production by ILC2 (after exogenous IL-33 stimulation)[27]. However, no direct relationship was provided to link the ILC2 and atherosclerosis phenotypes[27]. Thus, the role of naturally occurring ILC2 and the mechanisms through which they may regulate atherosclerosis are still unknown.

Thus, the focus of this work is in defining how ILC2 respond to hypercholesterolaemia, and how atherosclerosis develops in an environment where this cell type is absent. Our results show that ILC2 control the development of atherosclerosis, in part through production of type 2 cytokines.

## Results

### Characterization of ILC2 in atherosclerosis-prone mice.
We first addressed the frequency of ILC2 in atherosclerosis-susceptible apolipoprotein e-deficient ($Apoe^{-/-}$) mice fed normal chow diet. A typical overview of a transverse section of the aorta reveals several aortic and para-aortic structures (Supplementary Fig. 1A) where ILC2 may accumulate. Those structures include the atherosclerotic plaque, the aortic adventitia and associated tertiary lymphoid structures (ATLO), the para-aortic lymph nodes (PaLN), the para-aortic adipose tissue (PaAT) and fat-associated lymphoid clusters (FALCs). We therefore examined and quantified the presence of ILC2 in each of those structures using flow cytometry and immunofluorescence. Our analyses first revealed the presence of ILC2 (Lin$^-$ ICOS$^+$ CD25$^+$ CD127$^+$) in PaLN and PaAT of chow-fed $Apoe^{-/-}$ mice (Fig. 1a), which is consistent with the previously reported presence of ILC2 in secondary lymphoid organs (GATA3$^+$ ICOS$^+$CD3$^-$ cells in Supplementary Fig. 1B,C) and their tropism for adipose tissue (for example, peri-gonadal WAT) (Fig. 1a)[28]. The percentage of ILC2 among CD45$^+$ cells in PaAT was smaller than in peri-gonadal WAT (Fig. 1b), but was substantially higher than the percentage of ILC2 in PaLN (Fig. 1b), and mesenteric lymph nodes (MLN) (Fig. 1b) of the same animals. Supplementary Fig. 1D shows the absolute number of ILC2 recovered from different locations in > 20-week-old $Apoe^{-/-}$ mice.

The phenotype, activation state and function of ILC2 may change dependent on the tissue where they reside and the cytokine microenvironment[29–31]. We found that Lin$^-$ ICOS$^+$ CD25$^+$ CD127$^+$ ILC2 of peri-gonadal WAT (GWAT) were mostly KLRG1$^+$ST2$^+$ (Fig. 1a,b) and were comparable to natural ILC2 (ref. 29), whereas ILC2 of MLN and PaLN were in large majority KLRG1$^+$ST2$^-$ or ST2$^{low}$ (Fig. 1a,b), similar to a recently described population of inflammatory ILC2 with reduced ability to produce IL-5 and IL-13 (ref. 29). Interestingly, the ILC2 population in para-aortic fat differed significantly from that of peri-gonadal fat, and comprised a population expressing low amounts of ST2 on their surface (Fig. 1c). The number of PaAT ILC2 remained relatively constant during aging (Supplementary Fig. 1E). ILC2 were also found in ATLO of 80-week-old $Apoe^{-/-}$ mice with advanced atherosclerosis (Supplementary Fig. 2A,B).

Recent studies showed that inflammatory stimuli promote the formation of FALCs within WAT[32]. FALCs have been detected mostly in peri-gonadal, mesenteric and mediastinal WAT, with the pericardium accumulating a substantial number of clusters[32]. However, whether FALCs may also accumulate in the para-aortic region is still unknown. Given the role of inflammation in FALC formation, we reasoned that those clusters may be more prevalent in old animals (for example, 80 weeks). Indeed, we detected FALCs in the para-aortic WAT of both old WT and $Apoe^{-/-}$ mice (Fig. 1d,e, Supplementary Fig. 1A). As reported for other locations, para-aortic FALCs were rich in CD3$^+$ T cells, B220$^+$ B cells, CD138$^+$ plasma cells (Fig. 1f), PNA$^+$Ki67$^+$ germinal centre-like B cells and Foxp3$^+$ Tregs (Supplementary Fig. 2C), accumulated a few ILC2 (Fig. 1g), no follicular dendritic cells (CD35 staining in Fig. 1f) and were supplied with blood vessels, lymph vessels, high endothelial venules and ERTR7$^+$ conduits (Supplementary Fig. 2C). The number and size of peri-aortic FALCs were significantly greater in $Apoe^{-/-}$ mice compared to WT mice (Fig. 1e), supporting a role for vascular inflammation in promoting para-aortic FALC formation.

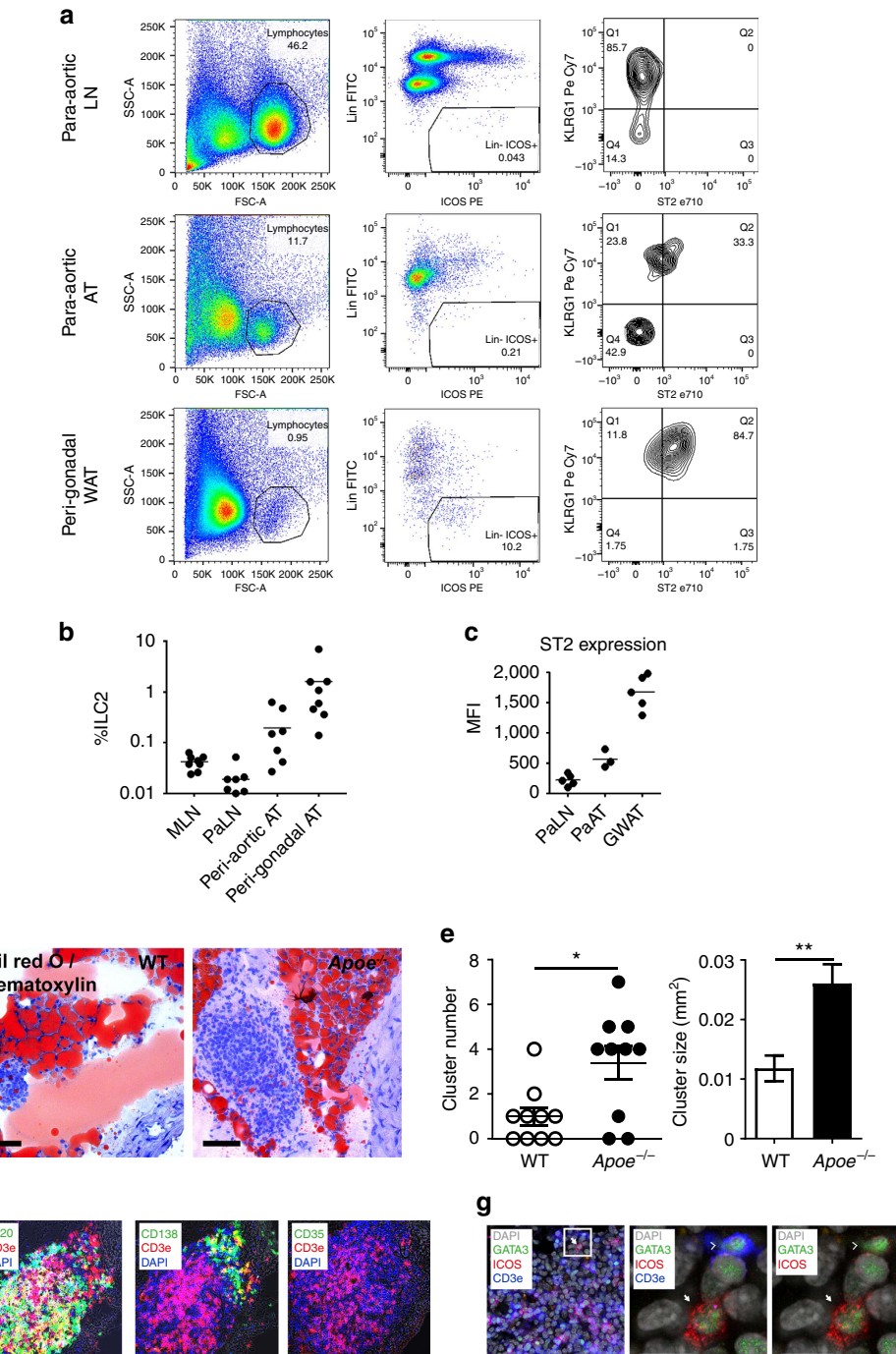

**Figure 1 | ILC2 in para-aortic adipose tissue and FALCs of atherosclerotic mice.** Flow cytometric analysis (**a**) and quantification (**b,c**) of ILC2 present in the para-aortic adipose tissue (AT) show a phenotype similar to KLRG1$^{hi}$ ST2$^-$ ILC2 (iILC2-like) in the lymph node in contrast to KLRG1$^+$ ST2$^+$ ILC2 (nILC2-like) prevalent in the peri-gonadal WAT. Mean fluorescence intensity (MFI) for ST2 expression in ILC2 is shown in **c**. Bars represent mean values. (**d**) Oil red O and Haematoxylin show that FALCs are present in the para-aortic AT of aged $Apoe^{-/-}$ mice (78–80 weeks of age $n = 10$, scale bar 100 μm) (**e**) and that $Apoe^{-/-}$ FALCs are greater in both number and size compared to WT (**f**). Immunoflorescence staining demonstrates that para-aortic FALCs are rich in CD3$^+$ T cells, B220$^+$ B cells and CD138$^+$ plasma cells. CD35$^+$ follicular dendritic cells were absent from these structures (scale bar 50 μm). Additionally, para-aortic FALC-resident CD3$^-$ GATA3$^+$ ICOS$^+$ ILC2 cells were also detected (**g**, scale bar 10 μm). Representative images shown. Graph data points represent individual mice. Statistical significance was determined by Mann–Whitney $U$-test.

Thus, besides their presence in WAT and secondary lymphoid organs, ILC2 are also present in para-aortic fat of athero-prone mice, where they display an inflammatory phenotype, distinct from the natural ILC2 phenotype of distant WAT and more similar to the inflammatory phenotype of lymph node ILC2.

**High fat feeding alters ILC2 numbers and cytokine production.** Mice fed a defined HFD for a period of weeks to months develop accelerated atherosclerosis. We therefore hypothesized that high fat feeding may alter the accumulation and function of ILC2 systemically, and observed the ILC2 populations in the bone marrow (BM), spleen and peripheral lymph nodes of low-density

lipoprotein receptor-deficient ($Ldlr^{-/-}$) mice, a second athero-sclerosis-susceptible strain, that had been maintained on HFD for 8 weeks. Flow cytometric analysis of ILC2 populations (Fig. 2a) demonstrated that although there was no difference in the proportion of precursor cells in the BM (Fig. 2b), the mature ILC2 were significantly under-represented (2–3-fold loss) in MLN and PaLN of mice maintained on HFD (Fig. 2b).

To examine any change in functional capability associated with the suppression of this population, Lin$^-$ ICOS$^+$ ILC2 cells were sorted from the spleens and MLN of conventional chow- and HFD-fed mice (purity > 95%, Supplementary Fig. 2D). Not only were fewer cells recovered from the organs of HFD mice (Supplementary Fig. 2E and consistent with Fig. 2b) but, during ex-vivo expansion with IL-7 and IL-33, they also secreted substantially less IL-5 and IL-13 (Fig. 2c). To confirm that the alteration of type II cytokine production occurred in vivo, we repeated the experiments and performed QPCR analysis on cell-sorted ILC2 isolated from the spleens of $Ldlr^{-/-}$ mice that had been maintained on chow or HFD for 8 weeks. ILC2 were also cell-sorted from the aortas (two pools of three mice each) and GWAT for comparison (Fig. 2d). QPCR analysis indicated a

significant decrease of GATA3 and IL-13 transcripts (Fig. 2d) and a similar trend observed with IL-5 (Fig. 2d), in spleen-derived ILC2 of mice on HFD compared to chow diet. Interestingly, GWAT-derived ILC2 showed no significant change, whereas aorta-derived ILC2 tended to upregulate their expression of GATA3, IL-5 and IL-13 after HFD (Fig. 2d). This is a strong indication that HFD differentially alters ILC2 phenotype in the periphery, and that continuous production of type II cytokines by aortic-ILC2 may be critical to maintain a counter-regulatory pathway, and limit the progression of aortic inflammation in face of a sharp decline of type II cytokine production by peripheral ILC2.

**Expansion of ILC2 reduces atherogenesis.** Similarly to others[26], we hypothesized that reconstitution of ILC2 cells by treating with IL-2 during HFD would replenish an atheroprotective environment. To minimize off-target effects of IL-2 on other CD25-expressing cells such as Tregs, we used T- and B-cell-deficient $Apoe^{-/-}/Rag2^{-/-}$ mice maintained on HFD for 8 weeks. The mice received three weekly injections of IL-2/Jes6-1

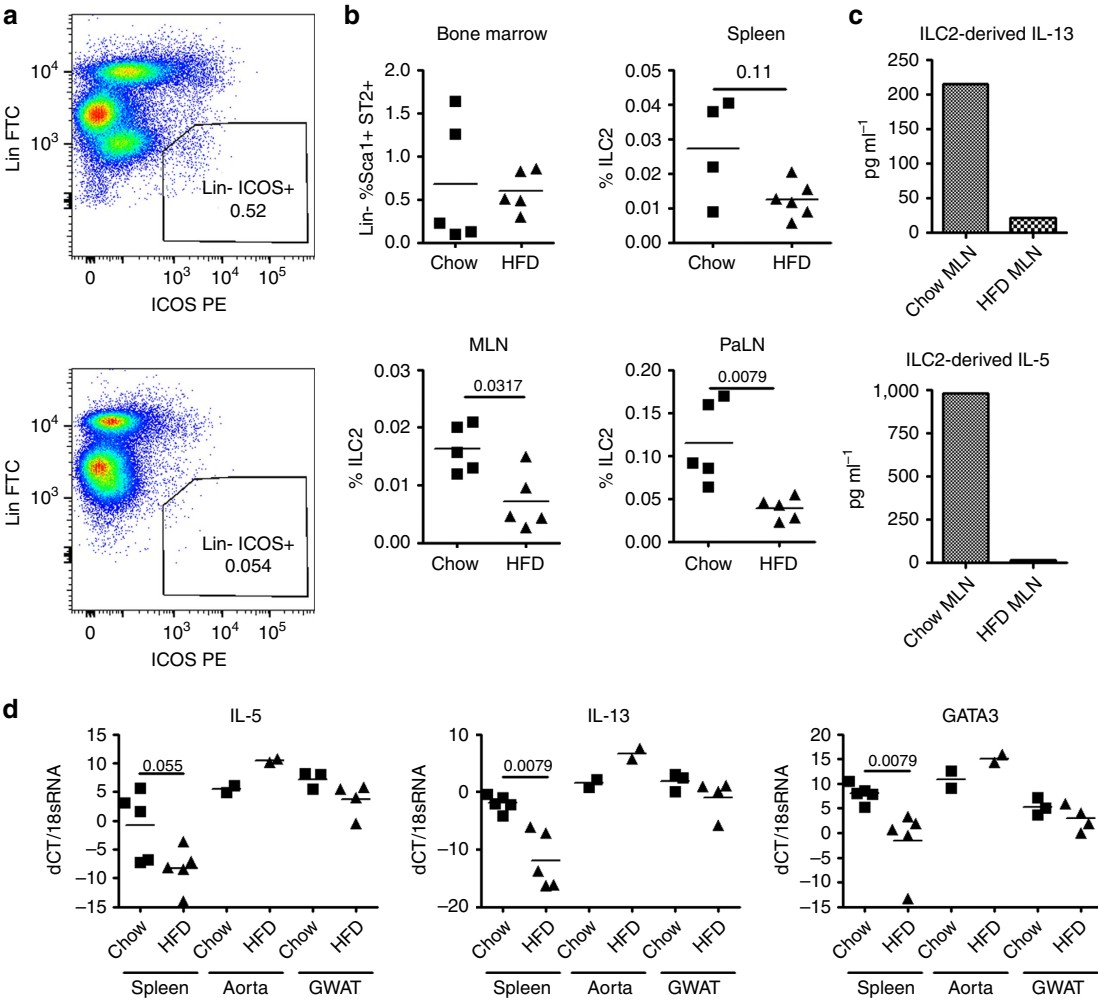

**Figure 2 | ILC2 populations are decreased during high fat diet.** Spleen, bone marrow (BM), mesenteric lymph node (MLN, shown) and para-aortic lymph node (PaLN, shown) from $Ldlr^{-/-}$ mice maintained on high fat diet (HFD) for 8 weeks were analysed for ILC2 populations by flow cytometry (**a**). While BM resident ILC2 were unchanged (top left) a downward trend in splenic ILC2 and a statistically significant decrease in ILC2 was observed in MLN and PaLN (**b**). ELISA analysis of IL-5 and IL-13 in the supernatants of sorted and cultured ILC2 ($1 \times 10^5$ cells per well) also demonstrated a decrease in cytokine secretion (**c**). QPCR on sorted splenic, aortic and GWAT ILC2 from $Ldlr^{-/-}$ mice maintained on chow or high fat diet (HFD) for 8 weeks. Data from aortic ILC2 represent two pools of three mice each. Each other square or triangle represents data from one separate mouse (**d**). Statistical significance was determined by Mann–Whitney U-test.

complex, which increases IL-2 biological activity[33], for the duration of the experiment. Following this treatment, flow cytometry demonstrated ILC2 populations were significantly expanded in spleen and BM compared to PBS-treated controls (Fig. 3a). In addition to ILC2 expansion in peripheral lymphoid tissue, clusters of ICOS[+] KLRG1[+] (Fig. 3b) ILC2 cells were observed in the adventitia of the aorta adjacent to the aortic sinus by immunofluorescence. The adventitia has been suggested as a source of precursor cells, which may influence plaque architecture[34] and the presence of expanded ILC2 in this tissue may suggest a direct localized effect. Whether these ILC2 have expanded *in situ* (as recent publications may suggest[35]) or have migrated into the tissue from the periphery remains to be investigated.

Further phenotypic changes occurred during this IL-2/Jes6-1 treatment, namely an expanded population of IL-5[+] ILC2 (Lin[−] ICOS[+]), associated eosinophilia and decreased CD11b[+] Ly6G[−] Ly6C[hi] inflammatory monocytes (Supplementary Fig. 3A).

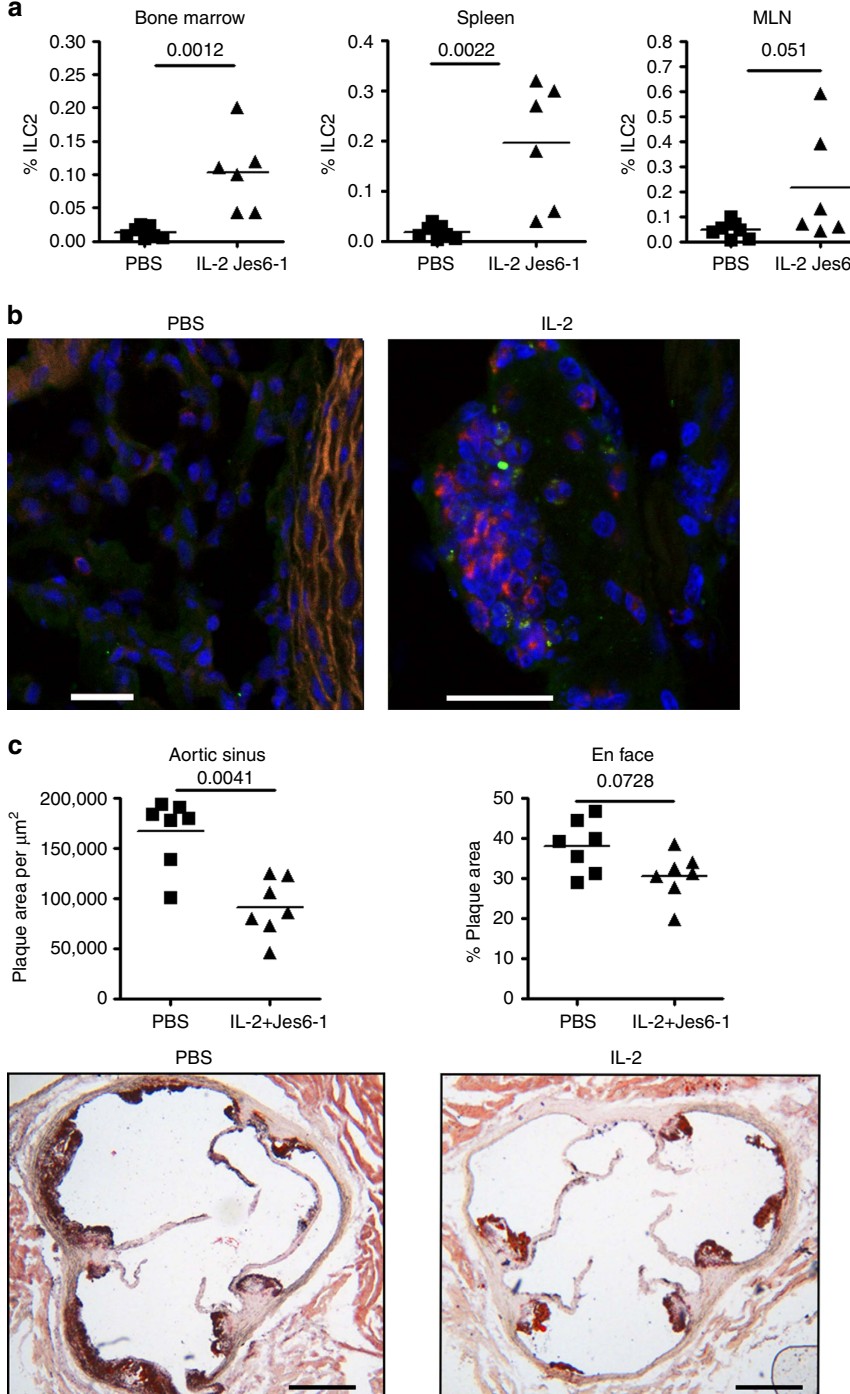

**Figure 3 | ILC2 expansion reduces atherosclerosis in *Apoe*[−/−]/*Rag2*[−/−] mice.** IL-2/Jes6-1 complexes can expand ILC2 in the bone marrow and spleen of *Apoe*[−/−]/*Rag2*[−/−] mice (**a**) as well as inducing clusters of ICOS[+] KLRG1[+] cells (**b**, ICOS Green, KLRG1 Red, scale bar 25 μm) adjacent to the aortic sinus. IL-2/Jes6-1 treatment significantly decreases plaque area in the aortic sinus compared to vehicle alone (**c**, scale bar 270 μm). Graph data points represent individual mice and statistical significance was determined by Mann–Whitney *U*-test.

There was also an increase in splenic NK cells (Supplementary Fig. 3A). This treatment resulted in a significant decrease in the area of atherosclerotic plaques at the aortic sinus (Fig. 3c) and a similar trend was also observed at the aortic arch although this was not statistically significant. In contrast to a recent study where IL-2/Jes6-1 treatment of $Ldlr^{-/-}/Rag2^{-/-}$ mice led to a substantial reduction of plasma VLDL-cholesterol levels[26], our treatment of $Apoe^{-/-}/Rag2^{-/-}$ mice with IL-2 was successful in reducing the effect of HFD on the progression of atherosclerosis, without any change in plasma lipid levels (Supplementary Fig. 3B). However, as reported in previous work[26], IL-2 treatment was not without a number of off-target effects, the most common of which was an increase in splenic fibrosis in the IL-2 treated group, which was severe in some mice (Supplementary Fig. 3C). Additionally, weight gain upon HFD feeding was slower in the group receiving IL-2 compared to controls (Supplementary Fig. 3D).

**Selective genetic ILC2 ablation exacerbates atherosclerosis.** Artificial expansion of ILC2 does not inform about the true role of the endogenous ILC2 population that develops during the course of atherogenesis. To allow the specific depletion of ILC2 in an otherwise replete immune system, $Staggerer/Ror\alpha^{Flox}$-$Cd127^{Cre}$ mice (which are selectively deficient in ILC2, hereafter known as ILC2$^{KO}$ (ref. 36)) were used as donors in a BM transplant model into atherosclerosis prone $Ldlr^{-/-}$ recipients. Given recent observations that in steady-state conditions tissue resident ILC2 are not replenished from circulating ILC2 (ref. 35), we validated the ability for BM ILC2 from Thy1.1 congenic mice to reconstitute lymphatic and tissue compartments. Thy1.2$^+$ recipient mice were irradiated and reconstituted with Thy1.1$^+$ BM. Following a 4-week recovery period, spleen, MLN and GWAT were collected and the proportion of Thy1.1$^+$ donor ILC2 was determined by flow cytometry. We found that donor ILC2$^{Thy1.1}$ fully reconstituted the lymphatic compartments (0% Thy1.2$^+$ ILC2) as well as the majority of the GWAT tissue resident ILC2 (5% Thy1.2$^+$) (Fig. 4a). Therefore, BM transplants are an effective method for replacing host with donor BM-derived ILC2. This was then repeated using either WT or ILC2$^{KO}$ BM followed by recovery and HFD for 9 weeks. To ensure the BM graft was effective and very few endogenous ILC2 remained, IL-33 was given to recipients 24 h before organ collection. Subsequent flow cytometry analysis demonstrated that recipients of ILC2$^{KO}$ BM had significantly decreased ILC2 in BM and peripheral MLN compared to ILC2$^{WT}$ recipients (Fig. 4b). Quantification of serum cytokines also demonstrated decreased IL-5 and IL-13 (Fig. 4c). Moreover, gene expression analysis on aorta and PaAT confirmed a substantial reduction of IL-5 and IL-13 expression in tissues recovered from ILC2$^{KO}$ mice (Fig. 4d). The extent of lipid accumulation in aortas of ILC2$^{KO}$ mice was significantly increased in the aortic arch (Fig. 4e) and aortic sinus (Fig. 4f) compared to ILC2$^{WT}$ recipients, despite no change in plasma lipid levels (Supplementary Fig. 4A).

As discussed above, ILC2 are a potent source of the type II cytokines IL-5 and IL-13, which are known to be atheroprotective through differing mechanisms. It was therefore logical to investigate whether the increase of disease severity in the ILC2$^{KO}$ model could be accounted for by deficiencies in these pathways. The potential expansion and maintenance of B1a B cells and associated increase in natural IgM antibodies by ILC2-derived IL-5 were investigated by flow cytometry in ILC2$^{KO}$ BM recipients. Although pooled results from a number of experiments suggested that B1a B cells were less abundant in the spleen and MLN of ILC2$^{KO}$ recipient mice (Supplementary Fig. 4B), there was considerable variation between biological replicates and

not all repeats followed the same trend. Furthermore, there was no associated decrease in natural IgM isotypes in the serum of ILC2$^{KO}$ recipient mice after 8 weeks of HFD (Supplementary Fig. 4B) or difference in plaque IgM deposits (Supplementary Fig. 4C). Thus, changes in B1a B-cell subset are unlikely to account for the effect of ILC2 deletion on atherosclerosis although this remains to be clarified.

**ILC2 alter plaque composition.** To examine if ILC2 deletion impacts immune cell accumulation and activation *in vivo*, we analysed plaque composition. The number of CD3$^+$ T cells in lesions of ILC2$^{KO}$ recipients was reduced (Supplementary Fig. 5A) indicating that the increase in lesion size was unlikely to be driven by T-cell activation. However, immune-fluorescent labelling of MOMA2$^+$ myeloid cells in the aortic sinus revealed a larger lipid-containing core of foam cells in the absence of ILC2 (Fig. 5a). Interestingly, the expression of Arg1 was significantly decreased (Fig. 5b). There was no difference in the proportion of α-smooth muscle actin-expressing cells in plaques or the deposition of collagen throughout the plaque detected by Sirius red staining (Supplementary Fig. 5B). It is usual for larger, more advanced plaques in this model to contain more collagen deposits, and this absence of increased collagen deposition in the larger plaques of ILC2$^{KO}$ mice coupled with less Arg1 expression might indicate disrupted tissue repair mechanisms. The macrophage phenotype was therefore further investigated by flow cytometry. Here, we observed a significant decrease in CD11b$^+$ F4/80$^+$ Arg1$^+$ and CD11b$^+$ F4/80$^+$ Arg1$^+$ iNOS$^+$ macrophage population in the aorta and peri-aortic adipose tissue of ILC2$^{KO}$ mice and an expansion of CD11b$^+$ F4/80+ iNOS$^+$ macrophages (Fig. 5c). This shows that, although ILC2 are a rare population of cells, in mouse models of atherosclerosis they perform a critical role in preventing plaque development and their ablation alters macrophage phenotype and increases disease severity.

**ILC2-derived IL-5 and IL-13 are required for atheroprotection.** We designed reconstitution experiments to address the specific roles of ILC2-derived IL-5 or IL-13 in the control of atherosclerosis. BM transplantation experiments were performed in $Ldlr^{-/-}$ mice which received mixed BM transplants of ILC2$^{WT}$, ILC2$^{KO}$, 80% ILC2$^{KO}$ with 20% IL-5$^+$ (ILC2-deficient mice reconstituted with IL-5 sufficient ILC2) or 80% ILC2$^{KO}$ with 20% IL-5$^{KO}$ (ILC2-deficient mice reconstituted with IL-5-deficient ILC2; 80% of all other cell types are still capable of IL-5 production). After recovery, mice were put on HFD for 8 weeks. Reproducing the original observation, ILC2$^{KO}$ mice showed increased atherosclerosis of the aortic arch (Fig. 6a). Additionally, ILC2$^{KO}$ mice reconstituted with IL-5$^+$ ILC2 did not develop increased atherosclerosis compared to ILC2$^{WT}$ mice (Fig. 6a), further supporting the requirement for a competent ILC2 population to limit atherogenesis. Crucially however, recipients of ILC2$^{KO}$/IL-5$^-$ BM, which are replete with IL-5-deficient ILC2, developed severe atherosclerosis comparable to the full ILC2 knockout condition (Fig. 6a). However, absence of ILC2-derived IL-5 did not alter lesion size in the aortic sinus (Supplementary Fig. 5C), suggesting the involvement of other pathways. Therefore, complementing the observations with ILC2 sourced IL-5, we examined the function of ILC2-derived IL-13 in a separate set of experiments by reconstituting BMT recipients with 80% ILC2$^{KO}$ and either 20% IL-13$^+$ or 20% IL-13$^{KO}$. As was observed with the ILC2-specific IL-5 deficiency, the inability of ILC2 to produce IL-13 significantly increased atherosclerosis in the aortic arch (Fig. 6a). Furthermore, there was a significant increase in lesion size in the aortic sinus of ILC2$^{KO}$ IL-13$^-$

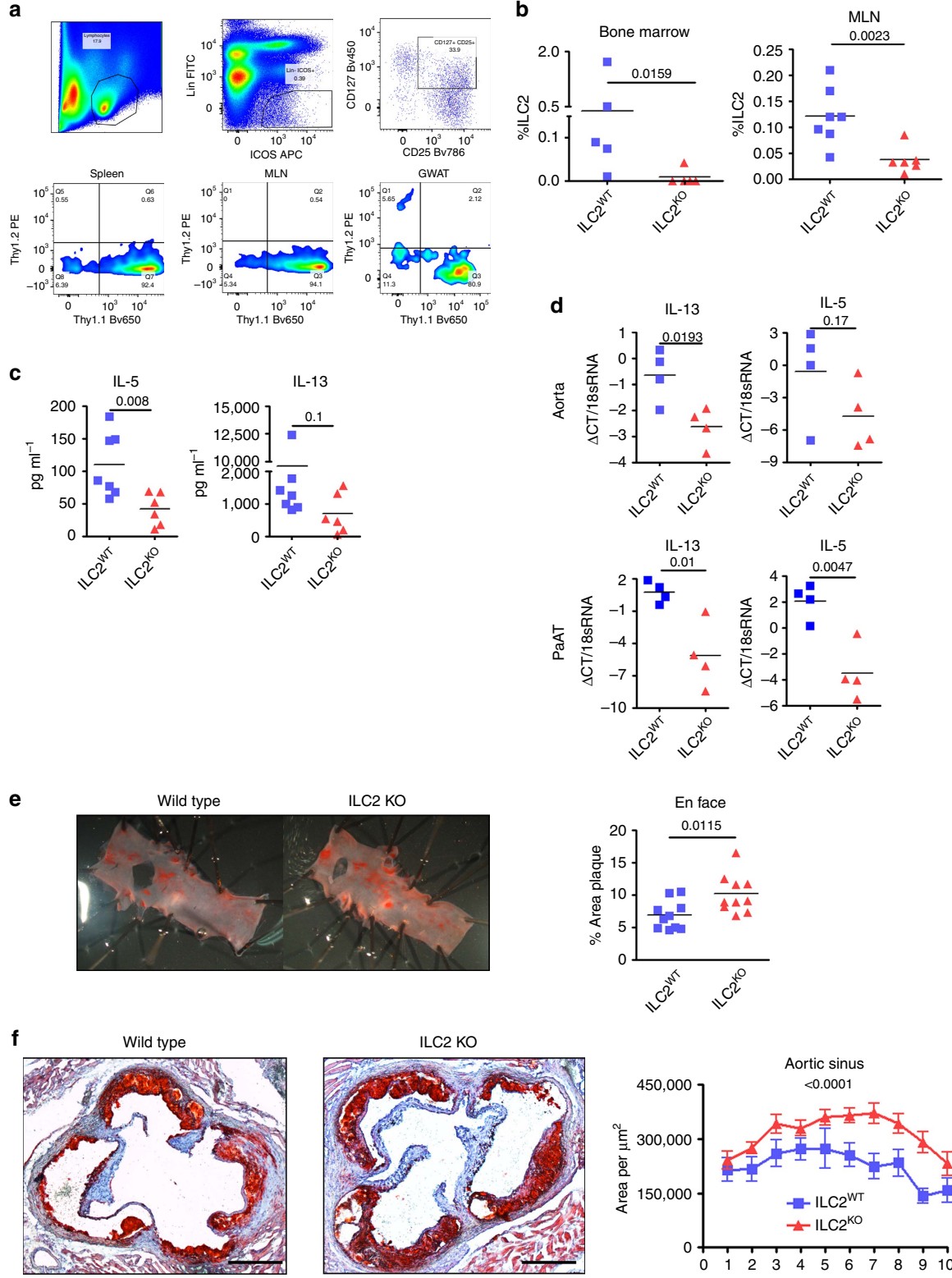

**Figure 4 | Genetic deletion of ILC2 exacerbates atherosclerosis in *Ldlr*$^{-/-}$ mice.** During bone marrow transplant, ILC2$^{Thy1.1}$ fully restore Spleen MLN and GWAT resident compartments in Thy1.2$^+$ mice (**a**). Irradiated *Ldlr*$^{-/-}$ mice received bone marrow from Stagger/Rorα$^{Flox}$-CD127$^{Cre}$ (ILC2$^{KO}$) or Stagger/Rorα$^+$-CD127$^{Cre}$ (ILC2$^{WT}$) donor mice before being maintained on HFD for 8 weeks (**b**). Intra-peritoneal injections of IL-33 24 h before organ collection demonstrated recipients of ILC2$^{KO}$ BM had decreased Lin$^-$ ICOS$^+$ Sca1$^+$ ST2$^+$ILC2 in bone marrow and Lin$^-$ ICOS$^+$ CD25$^+$ in peripheral MLN compared to ILC2$^{WT}$ recipients. Serum levels of IL-5 and IL-13 were also decreased (**c**), as was expression of IL-5 and IL-13 transcripts in aorta and PaAT (**d**). Oil Red O quantified atherosclerotic lesions at the aortic arch and aortic sinus (**e** and **f** respectively, representative images shown) indicated increased plaque size (all surface of intimal lesion is taken into account) in ILC2$^{KO}$ recipients compared to ILC2$^{WT}$ controls for both sites. Graph data points represent individual mice and statistical significance was determined by Mann–Whitney *U*-test. Scale bars: 270 μm.

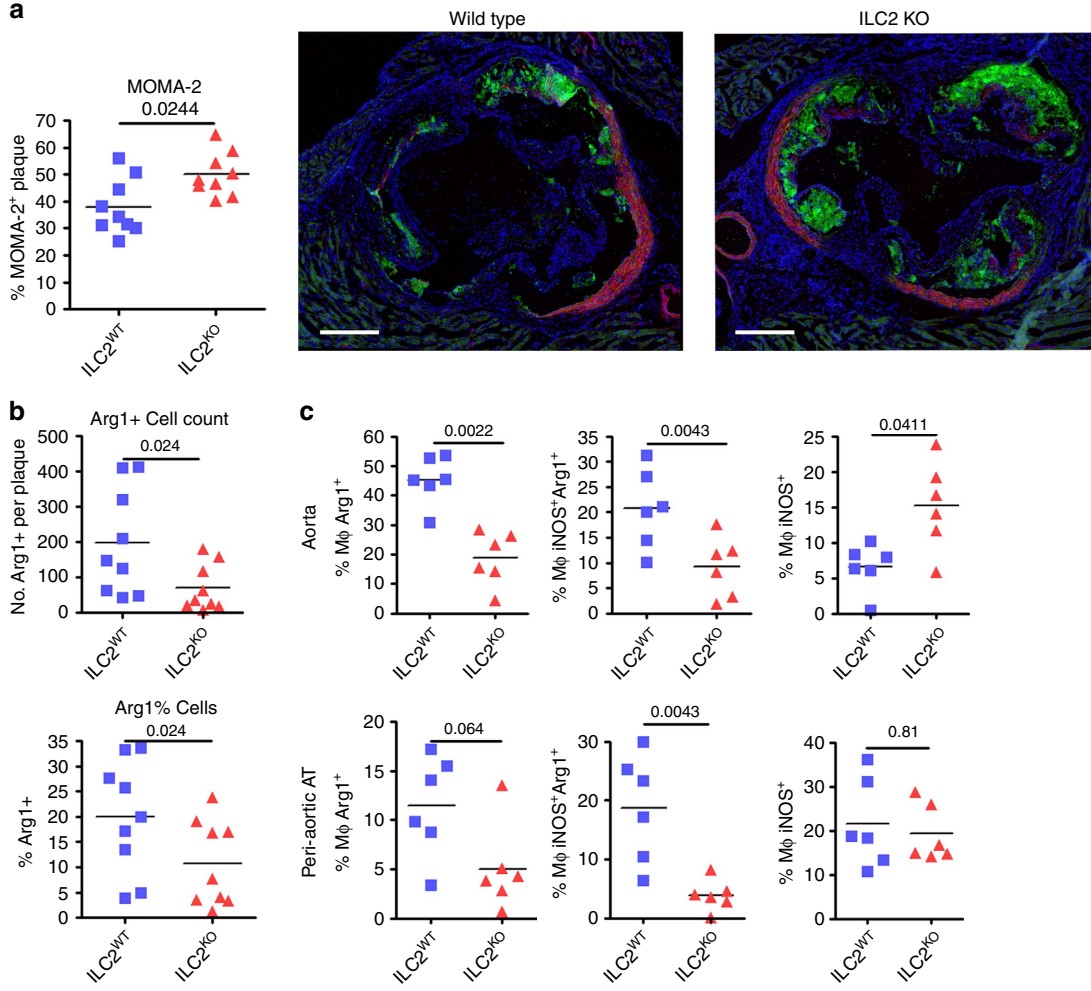

**Figure 5 | ILC2 alter the plaque composition.** Phenotypic changes in the enlarged plaque (**a**) favour increased MOMA2+. Proportional and numerical decrease in Arginase1 (Arg1) expressing cells (**b**). Flow cytometry of CD11b+ F4/80+ aorta resident macrophages indicated bias towards reduced Arg1 and increased iNOS expression (**c**). Representative micrographs of MOMA2+ staining are shown. Graph data points represent individual mice and statistical significance was determined by Mann–Whitney *U*-test. Scale bars: 270 μm.

recipients (Fig. 6b). Although no significant change in MOMA2+ expression was detected (Fig. 6c), Arg1 expression was decreased (Fig. 6c) as previously observed in ILC2KO recipients. We also found a significant reduction of collagen deposition in lesions of ILC2KO IL-13− recipients (Fig. 6d), indicating impaired vascular healing. ILC2-derived cytokines (IL-5 and IL-13) are therefore vital components controlling the progression of atherosclerosis, particularly IL-13 which may alter macrophage phenotype, and its absence leads to larger and potentially more vulnerable plaques.

## Discussion
Here, we show that ILC2 constitute a major atheroprotective cell type. High fat feeding reduces the frequency of ILC2 in the periphery and profoundly alters their protective phenotype, concomitant with an acceleration of atherosclerosis. Using mice specifically deficient in ILC2, we show that endogenous ILC2 perform a central role in controlling the progression of atherosclerosis and this effect is in part dependent on ILC2-derived IL-5 and IL-13. Remarkably, production of IL-5 and IL-13 by other cell types is unable to compensate for the lack of those ILC2-derived cytokines, particularly IL-13, and their atheroprotective effects. IL-5-dependent atheroprotection was limited to the thoracic aorta and could not be attributed to

changes in macrophage phenotype or B1-dependent natural IgM production. IL-13-dependent atheroprotection was associated with important changes in collagen deposition and macrophage phenotype, suggestive of alternative activation. However, the direct links between changes of macrophage phenotype and atheroprotection were not addressed. Future studies should try to understand the differential impact of HFD on peripheral versus aortic ILC2, and define their distinct contributions to limiting vascular inflammation and atherosclerotic lesion development.

Previous studies suggested a potential role for ILC2 in the modulation of atherosclerosis[23,26]. However, those studies used immune-compromised animals and relied on non-physiological exogenous and chronic administration of cytokines (that is, IL-2 and IL-25) that are not specific for the ILC2 population, and that can promote ILC2-independent immune responses. Moreover, those studies failed to provide any direct evidence of the involvement of ILC2 in atherosclerosis and were confounded by profound alterations of hepatic and lipid metabolism[26] following chronic exogenous cytokine administration. We also found that chronic administration of IL-2/anti-IL2 complexes in immune-compromised animals reduced atherogenesis but in agreement with previous findings, the effect was associated with several adverse side effects. Most probably, the amount of IL-2/IL-2 mAb used in murine models is not physiologically relevant and further

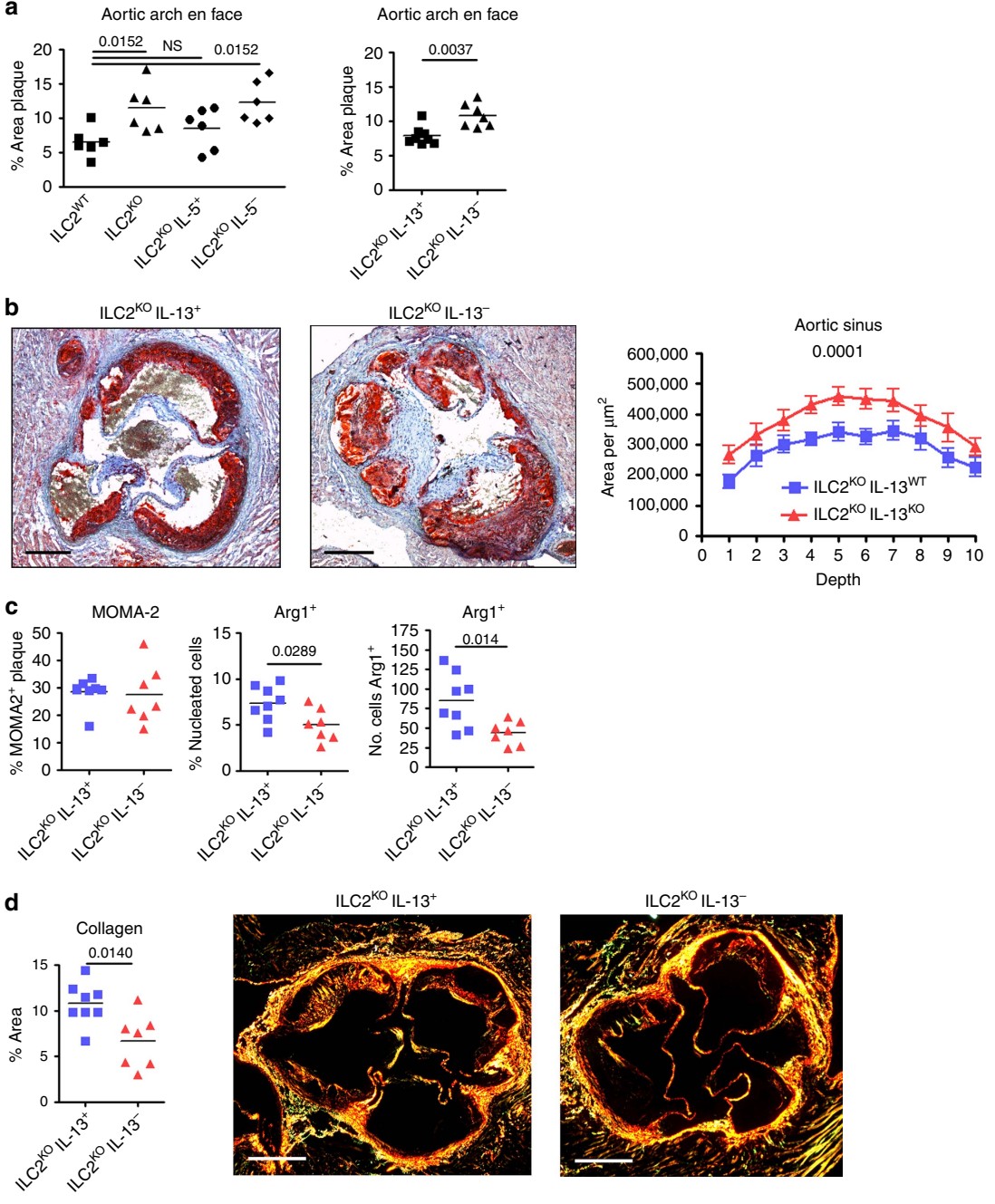

**Figure 6 | ILC2 derived IL-5 and IL-13 are required to reduce atherosclerosis.** $Ldlr^{-/-}$ recipients of mixed bone marrow transplants of ILC2$^{WT}$, ILC2$^{KO}$, 80% ILC2$^{KO}$ with 20% IL-5$^+$ or 80% ILC2$^{KO}$ with 20% IL-5$^{KO}$, 80% ILC2$^{KO}$ with 20% IL-13$^+$ 80% ILC2$^{KO}$ with 20% IL-13$^-$ were maintained on HFD for 8 weeks. Atherosclerosis in the aortic arch was quantified after Oil Red O staining, multiple comparison using Kruskal–Wallis yielded a *P* value of 0.03 (**a**). Representative images and quantification of atherosclerotic lesion size in aortic sinus of ILC2$^{KO}$ IL-13$^-$ and ILC2$^{KO}$ IL-13$^+$ recipients (**b**). Proportional and numerical decrease of Arg1$^+$ cells in aortic sinus of ILC2$^{KO}$ IL-13$^-$ recipients (**c**). Reduced collagen deposition in lesions of ILC2$^{KO}$ IL-13$^-$ recipients (**d**). Graph data points represent individual mice and statistical significance was determined by Mann–Whitney *U*-test or two-way ANOVA where appropriate. Scale bars: 270 μm.

work is required to titrate the dose of cytokine required to protect against atherosclerosis without inducing undesirable side effects.

It is interesting to note, however, that low dose IL-2 therapy is reported to be safe in humans and has been successful in several clinical trials of immune-mediated diseases (reviewed in ref. 37) where exogenous IL-2 was used to expand Tregs. Remarkably, low-dose IL-2 in humans also increases the production of IL-5 (ref. 38) and this is attributed to the dose-dependent expansion of ILC2.

Thus, augmentation of ILC2 and ILC2-derived IL-5 or IL-13 on top of Treg expansion might constitute a potentially attractive double-hit therapy to limit accelerated atherosclerosis.

## Methods

**Mice.** All work was conducted under UK Home Office project license regulations after approval by the Ethical Review Committee of the University of Cambridge.

Mice used in this investigation were $Ldlr^{-/-}$ (Jackson Labs 002207), $Apoe^{-/-}/Rag2^{-/-}$ (Jackson Labs), and $IL5^{-/-}$ were from Manfred Kopj

(ETH Zurich). IL-13$^{gfp/gfp}$ (ref. 12) and Staggerer/Ror$\alpha^{Flox}$-CD127$^{Cre}$ mice were from A. McKenzie and H. Rodewald[36,39]. All mice were on the C57/Bl6 background apart from Thy1.1$^+$ and Thy1.2$^+$ mice which were Balb/c.

**Bone marrow transplants.** Eight-week-old female recipient mice were maintained over night with Baytril before irradiation with two doses of 5.5 Gy (separated by 4 h) followed by reconstitution with $1 \times 10^7$ sex-matched donor BM cells. Mice were then maintained on Baytril for a 4-week recovery period before organ collection (Thy1.1$^+$/Thy1.2$^+$) or fed either normal chow (SAFE diet 105) or Western High Fat Diet (Dietex, FAT 21%, Cholesterol 0.15%) for 8–9 weeks (ILC2$^{KO}$, ILC2$^{KO}$ IL-5$^{KO}$ and ILC2$^{KO}$ and IL-13$^{KO}$ experiments).

**In vivo ILC2 expansion.** To expand ILC2 in Apoe$^{-/-}$/Rag2$^{-/-}$, IL-2 (Preprotech) was complexed with monoclonal antibody Jes6-1 (Bio legend #503701) at a ratio of 5:1, incubating at 37 °C for 30 min (ref. 40). Mice received 1 μg of complex IL-2 three times a week for 8 weeks. During this period mice were maintained on HFD (Western RD).

**ILC2 isolation.** The ILC2 population was expanded in mice with three daily injections of IL-33 (1 μg) intraperitoneally before spleens and lymph nodes were collected. Cell suspensions were prepared for flow sorting following standard protocols. ILC2 cells were sorted from the Lineage-negative ICOS-positive population and maintained in vitro. Ex vivo sorted ILC2 cells were plated at a density of $5 \times 10^4$ cells per well in RPMI (10% FCS Pen/Strep 50 μM 2-ME) supplemented with IL-7 and IL-33 (10 ng ml$^{-1}$ each).

**Molecular biology.** RNA was extracted from ex vivo stimulated macrophages using RNEasy kit (Qiagen) following the manufacturer's instructions followed by cDNA synthesis of 1 μg total RNA using (Qiagen). For ILC2$^{KO}$ BMT aorta and PaAT single cell suspensions were produced and RNA extracted using RNEasy micro$^+$ kit (Qiagen) followed by cDNA synthesis using SMART v4 Ultra low input RNA kit (Clontec), with 11 cycles of PCR.

For ILC2 isolation and QPCR, 100 Lin$^-$ ICOS$^+$ CD25$^+$ ILC2 were directly sorted into SMART v4 lysis buffer (SMART v4 Ultra low input RNA kit, Clontec) and cDNA amplified as per instructions (14 cycles of PCR). For QPCR, a 1:50 dilution of cDNA pools was used with MESAGreen (Eurogentec) and cycled on a Lightcycler 480 (Roche). Primer sequences are as follows: GATA3 (For 5′-AAA GAA GGC ATC CAG ACC CG-3′ Rev 5′-TTG AAG GAG CTG CTC TTG GG-3′), IL-5 (For 5′-CAA GCA ATG AGA CGA TGA GGC-3′ Rev 5′-CCC ACG GAC AGT TTG ATT CTT C-3′), IL-13 (For 5′-TGT GTC TCT CCC TCT GAC CC-3′ Rev 5′-GAG GCC TAC ACA GAA CCC G-3′), Arg1 (For 5′-TGA AGA GCT GGC TGG TGT GGT-3′ Rev 5′-GCT TCC AAC TGC CAG ACT GTG GTC-3′), Fizz1 (For 5′-TCA AGG AAC TTC TTG CCA ATC-3′ Rev 5′-ACC CAG TAG CAG TCA TCC CAG-3′).

**Flow cytometry.** The following antibodies were used, brackets denote clone. Lineage cocktail comprises of CD3e (2c11), CD4 (RM4-5), CD19 (1D3), CD11b (M170) CD11c (N418) Gr1 (RB6-8C5) NK1.1 (PK136), FceRI (MAR-1) and Ter119, all FITC conjugated and used at a final concentration of 0.1 μg ml$^{-1}$. ILC2 panel: ICOS-APC (C398.4A), ST2-e710 (RMST2-2), CD127 e450 (A7R34), CD25$^-$ Bv786 (3C7) KLRG1-PE Cy7 (2F1) were diluted to 0.2 μg ml$^{-1}$ and incubated in the presence of 24G2 Fc receptor blocking. ILC2 were defined by flow cytometry as Lin$^-$ ICOS$^+$ CD127$^+$ CD25$^+$ KLRG1$^+$ ST2$^{variable}$. Macrophage phenotyping: CD11b$^+$ F4/80$^+$ Arg1$^+$/iNOS$^+$. CD11b-PE (M1/70 0.1 μg ml$^{-1}$), F4/80 Pacific Blue (BM8 0.1 μg ml$^{-1}$), iNOS Alexa647 (Polyclonal, Insight bio-technology 0.2 μg ml$^{-1}$) Arginase-1 (Polyclonal N-20, Santa Cruz Biotechnology 0.2 μg ml$^{-1}$), Chicken anti-Goat AF488 (ThermoFisher Scientific). Cells were washed and run on a LSR- Fortessa (BD Biosciences). Subsequent data were analysed with FloJo X analysis software (FreeStar Ashland, OR, USA).

**Histology and immunohistochemistry and morphometry.** Tissues were collected into 1% PFA overnight before washing into PBS. Quantification of atherosclerosis was performed using Oil red O staining as previously described[41]. Briefly, en face wholemount staining was performed on aorta cleaned of all peri-aortic adipose tissue and adventitia. A 0.5% working solution of Oil Red O (Sigma #O0625) dissolved in isopropyl alcohol was used, diluted to a working stock of 60% in distilled water. Aorta were rinsed with distilled water before a wash in 70% isopropyl alcohol. Whole aorta were immersed in working solution for 45 min followed by a wash in 70% isopropyl alcohol and 5 further washes in distilled water. For aortic sinus, OCT embedded frozen sections from PFA fixed hearts were air dried and washed in PBS followed by 60% isopropyl alcohol. Sections were then stained in freshly prepared Oil Red O working solution for 15 min, briefly washed in isopropyl alcohol before counterstaining with haematoxylin. Sections were then washed twice in Scott's solution before mounting in aqueous mountant (CC/Mount, Sigma). Plaque collagen content was determined using Sirius Red staining under polarized light. Lesion size/collagen was quantified using Fiji[42].

To delineate ATLOs and para-aortic FALCs, tissues were prepared as described[5,43]. Thus, 10 μm fresh frozen cross-sections were prepared and every

tenth serial section at 100 μm intervals was stained with Oil Red O/haematoxylin and number of para-aortic lymphoid clusters and their sizes were quantified. Very small clusters ($< 200$ μm in size) and clusters around perivascular nerve and ganglia were discarded from morphometry analyses. Immunofluorescence staining was performed as previously described[43], using marker antibodies. For ILC2 cells staining the following antibodies were used. Rabbit anti-human CD3e (F7.2.38, DAKO 2 μg ml$^{-1}$), rat anti-mouse GATA3 (KT77, Abcam 2 μg ml$^{-1}$), Armenian hamster anti-mouse ICOS (C398.4A, Biolegend 2 μg ml$^{-1}$), rat IgG2a isotype for GATA3 (R35-95, BD, 2 μg ml$^{-1}$) and Armenian hamster IgG Biotin (A19-3, BD, 2 μg ml$^{-1}$). DAPI was used to stain DNA. Secondary antibodies were used as previously described[44]. For macrophage phenotype staining the following antibodies were used: MOMA2 (Abserotech MCA519G, 2 μg ml$^{-1}$), rabbit anti mouse iNOS AF647 (Polyclonal, Insight biotechnology 5 μg ml$^{-1}$), goat anti mouse Arginase-1(Polyclonal N-20, Santa Cruz 5 μg ml$^{-1}$), and chicken anti goat AF488 (Thermofisher).

**Cytokine quantification.** In vitro expressed cytokine quantification was performed using IL-5 and IL-13 Duoset ELISA kit (R and D Systems) following the manufacturer's instructions. Serum IL-5 and IL-13 were detected by enhanced sensitivity CBA FlexSet (BD Biosciences), diluted 1:20.

**Statistical analysis.** Statistical analyses were performed using the GraphPad Prism 4 software (Graph Pad Software, San Diego, CA, USA). An unpaired t-test was used to analyse parametric data sets whereas for non-parametric data the Mann–Whitney U-test was applied. Tests performed and calculated two-tailed P-values are indicated in the individual figure legends.

**Data availability.** The data that support the findings of this study are available from the corresponding author on reasonable request.

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

## Acknowledgements

This research was supported by the Cambridge NIHR BRC Cell Phenotyping Hub. We thank Yuanfang Li (IPEK, LMU Munich) for para-aortic FALC morphometry. Funding bodies: British Heart Foundation RG/15/11/31593 and PG/15/99/31865 to Z.M.; ERC Starting Grant GA281164 to Z.M.; German Research Council YI 133/2-1 to C.Y.; HA 1083/15-4 to A.J.R.H.; and MO 3054/1-1 to S.M.

## Author contributions

S.A.N. contributed to the design of the experiments, conducted the experiments and was involved in writing the manuscript. S.M. and A.P.S. contributed to the design of the experiments and conducted the experiments. M.C., S.T., M.N., C.Y., D.H., L.L.K. and A.J.F. conducted the experiments. H.-R.R. contributed IL-7$^{cre}$ transgenic mice. J.A.W. and A.N.J.M. contributed to the design of the experiments, provided reagents and Rorα$^{sg/fl}$ IL7r$^{Cre}$ mice. C.J.B. and A.J.H. contributed to the design of the experiments and provided reagents. Z.M. contributed to the design of the experiments and supervised and was involved in the writing of the manuscript.

## Additional information

**Competing interests:** The authors declare no competing financial interests.

