## [Peer Review File · Nature Communications]

Reviewers' comments:

Reviewer #1 (expert in immunology)

Remarks to the Author:

In the manuscript by Newland et al, the authors show that ILC2 are present in in aortic adventitia, para-aortic adipose tissue and lymph nodes and that genetic ablation of ILC2 in *Ldlr*^{-/-} mice accelerated the development of atherosclerosis. This study is an extension of previous findings from other groups reporting the presence of ILC2 in aorta and their role in the development of atherosclerosis (Mantani et al, 2015; Engelbersten et al 2015). In comparison to these studies, the authors use a more selective approach of ILC2 depletion and show that accelerated development of atherosclerosis in *Ldlr*^{-/-} mice could be prevented by reconstitution with wild type but not *Il5*^{-/-} or *Il13*^{-/-} ILC2.

I have the following remarks:

In Figure 1A and relative text, the authors conclude that "the ILC2 population in para-aortic fat differed significantly from that of peri-gonadal fat, and comprised a substantially lower proportion of nILC2... favouring a predominance of iILC2 over nILC2". iILC2 cells are apparently not present in peripheral tissues in the steady state but can be elicited at many sites by helminth infection or IL-25 treatment (Huang and Paul, *Int Imm* 2016), while ST2 and KLRG1 expression differ *ex vivo* in different tissues. Thus, I would avoid the term iILC2 (unless phenotype analysis is performed after *in vivo* IL-25 treatment). Moreover, similar to previous publications (Mantani et al, 2015; Engelbersten et al 2015), in this study only a couple of ILC2 markers have been evaluated. Further phenotypic and functional characterization of ILC2 in the paraortic LN and AT compared to mesenteric LN and lung (or small intestine), where they represent a major ILC population, should be accomplished *ex vivo*. In addition to ICOS, KLRG1 and ST2, *ex vivo* staining of GATA3, Sca-1, CD25, IL17BR, CD127, c-kit, CD90 and cytokine expression (after *ex vivo* stimulation) should be performed.

In the text, pag 6 the authors state: "ILC2, which were not found in aortic adventitia of WT or very young *Apoe*^{-/-} mice (7 weeks old, not shown), increasingly accumulate a mixture of nILC2 (KLRG1+ST2+) and iILC2 (KLRG1^{high}ST2⁻ or low) in the adventitia of *Apoe*^{-/-} mice as disease progressed with age (Figure 1B), and were found in ATLO of old *Apoe*^{-/-} mice with advanced atherosclerosis (Figures S1D and S1E)". No data supporting this conclusion are shown. The authors need to show a quantitative and qualitative comparison of ILC2 frequencies, absolute numbers, phenotype and *ex vivo* cytokine production in paraortic LN and AT (compared to mLN and lung, or SI as control) in *Apoe*^{-/-} and WT mice at different time points during aging (or disease progression in *Apoe*^{-/-} mice). Quantitative analysis should be performed both by Flow cytometry and histology. Moreover, the authors show that "FALCs develop in the para-aortic WAT of old WT and *Apoe*^{-/-} mice", but "accumulated only a few ILC2". A quantification of ILC2 at this site and a comparison between numbers in old WT and *Apoe*^{-/-} mice would be again very helpful.

Pag 9: "To ensure the bone marrow graft was effective and very few endogenous ILC2 remained, IL-33 was given 24h prior to organ harvest": the reason why the authors inject IL-33 in this experimental setting is not very clear to me. The authors should show that recipient ILC2 are indeed efficiently depleted by irradiation in all organs. Although BM ILC2 seem efficiently depleted in the BM, in MLN there is only a 3 fold reduction of ILC2 (Fig 3B). What about ILC2 in paraortic LN and other organs? If recipient ILC2 remained, I cannot see how IL-33 could be of help as it would also activate and expand those ones.

Figure 4: the authors state that "ILC2 favor alternative activation of macrophages through production of IL-5 and IL-13". Most of these data have been generated *in vitro*, by stimulating splenic or peritoneal macrophages with ILC2 conditioned media. It would be important to validate these data in

vivo by analyzing macrophage phenotype in paraortic LN and AT of old Apoe^{-/-} mice and WT mice in the absence of ILC2 or of ILC2 lacking IL-5 or IL-13.

Minor comments:

It is not clear from Figure legends or Mat and Methods whether CD45 staining is performed or not. Gating on CD45⁺ Lin⁻ cells from tissue would be important to better define ILCs.

Legend to Figure 1G mentions a Figure 1G which is not there (and is also not commented in results) Figure 2A-B: the authors define ILC2 by using different markers in different compartments. Gata3 staining could be used.

Figure 2C: cytokine expression should be evaluated ex vivo, not after in vitro culture

Figure 2D: percentages and absolute numbers of ILC2 after IL-2 complex treatment should be quantified in the paraortic LN/adventitia and not only in BM and spleen

Legend to Figure 4 does not have C and D in the description.

Reviewer #2 (expert in atherosclerosis and immunology)

Remarks to the Author:

The authors show that depletion of ILC2 cells accelerates atherosclerosis, which can be rescued by adoptive transfer of wild-type, but not IL-13 or IL-5 deficient ILC2 cells. This is not entirely unexpected, because a 2015 ATVB paper showed similar results. There is another paper (ref 23) that also implicates ILC2 cells in atherosclerosis, but is not cited for this aspect.

1. It is surprising that IL5 and IL13 are both critically required in ILC2 cells as suggested in figure 4. No explanation is given for this.
2. The criticism of ref 26 is not fair. Similar to reference 26, the authors also use ILC2 cells expanded in Rag KO mice.
3. The authors looked for ILC2 cells in the para-aortic lymph nodes and the para-aortic adipose tissue, but not in the aorta proper. Why?
4. The image pair shown in figure 3C does not appear to be representative of the data.
5. ATLOs are really not observed in mice of the age studied here. They usually appear after 1 year of age.
6. In the introduction, the authors acknowledge that the role of Th2 cells in atherosclerosis is unknown (controversial), but in the methods they claim that type II cytokines are atheroprotective. This is highly controversial. For example, the Th2 disease asthma is known to exacerbate atherosclerosis.
7. The mouse model is not introduced. Why would CD127-Cre driven KO of RORa lead to specific depletion of ILC2 cells?
8. Figure 3: the lesion size is small. 9 weeks HFD is probably too short for Ldlr^{-/-} mice.
9. What are "normal conditions" for expanding ILC2 cells? The data shown in figure 4A seem to be from a highly contrived in vitro system.
10. The diets are not sufficiently defined: "normal Chow (manufacturer) or Western High 330 Fat Diet (Dietex)". Western diet is not really suitable in Ldlr^{-/-} mice unless cholesterol is also high.
11. Statistical concern: It appears that multiple comparisons were not corrected for.

Reviewer #3 (expert in atherosclerosis and immunology)

Remarks to the Author:

In the paper "ILC2 control the development of atherosclerosis in mice" Newland et al utilize a number of different approaches to demonstrate that ILC2 reside in various regions of the aorta, that they can produce IL-13 and IL-5, that they restrict development of atherosclerosis, and that they presumably

do so by influencing macrophage polarization. There have been at least two papers recently published that specifically investigated the role of ILC2 in atherosclerosis that arrived at the same conclusion (Engelbertson et al 2015; Mantani et al 2015). Other papers have implied this, while a protective role for IL-5 is known. Much of the data shown in the manuscript are therefore a repeat of those findings. That said, the authors do use a genetic model that specifically depletes ILC2, but the analysis using this model is somewhat thin and rudimentary. Moreover, the mechanistic analysis, that is, the suggestion that ILC2 protect against atherosclerosis by polarizing macrophages is very weak because the authors use only splenic and peritoneal macrophages and look at a few markers in vitro. As such, we have no idea whether ILC2 protect against atherosclerosis locally by acting on macrophages in vivo or whether that protection occurs elsewhere. As a very minor population, we also have no idea how much of a IL-5 and IL-13 source ILC2 are in atherosclerosis. Ultimately, the paper presents an incremental advance and fails to deliver a solid mechanistic analysis.

Specific comments

Major:

1. There have been two previous publications evaluating the role of ILC2 cells in atherosclerosis: Engelbertsen et al., 2015 and Mantani et al., 2015. The current manuscript spends a lot of time justifying the need for the current studies because the previous publications were flawed. Nonetheless, this manuscript actually approaches the question of ILC2 cells in atherosclerosis the exact same way, with the exception of an additional genetic model in Figure 3 and Figure 4. It is important to note that Figures 1 and 2 conform almost exactly with the data shown in the Engelbertsen and Mantani papers, significantly reducing the novelty of the current manuscript.
2. It was unclear which population of ILC2 cells the authors identified as the most important during the progression of atherosclerosis. The authors note that very few ILC2 cells are present in the para-aortic lymph nodes (LN) and para-aortic adipose tissue (AT), but show a considerable number of cells in the abdominal and thoracic aorta. The importance of each population is unclear, however, because despite the dominant ILC2 population in the aorta, the entire manuscript focuses on ILC2 cells in the mesenteric LN and para-aortic LN.
3. It is unclear how ILC2 cells increase in the aorta throughout the progression of atherosclerosis, but are reduced by high fat diet.
4. For the analysis in Fig. 2C, the authors state that after expansion of ILC2 cells ex vivo, less IL-13 and IL-5 were produced by ILC2 cells generated from HFD-treated mice. Fewer ILC2 cells were collected from these mice as well. Were equal numbers of cells plated ex vivo and did the authors confirm equal expansion of both ILC2 cell types? If there were fewer ILC2 cells from the HFD-treated mice, or the cells were less responsive to IL-7/IL-33, then the lower levels of IL-13 and IL-5 would result from the fact that there were fewer cells, not because they were qualitatively different in cytokine expression.
5. Overall the ex vivo data seem particularly weak. These data do not support a strong case for actions of IL-5 and IL-13 on macrophages as the driver behind reduced atherosclerosis with more ILC2 cells. Indeed, iNOS and TNF α , two factors strongly implicated in atherogenesis, remained unchanged with ILC2 treatment of macrophages. Moreover, these macrophage-associated effects were never evaluated in vivo. If the effects were indeed entirely related to changes in arginase-1 and fizz1 gene expression in macrophages, then the authors should show commensurate changes in plaque stability. Also, why would changes in arginase-1 and fizz-1 in macrophages affect how many macrophages accumulate within the plaque? Each of these factors needs to be addressed before macrophages can be implicated in the mechanism of reduced atherosclerosis in this model.

6. It is also unclear why the authors chose to do in vivo experiments with IL-5. IL-5 showed no great potential ex vivo and looked like it may have actually increased iNOS expression in peritoneal macrophages.

7. The conclusion to exclude a role for IgM seems a bit premature. There were fewer B1 cells in several tissues investigated, and it is possible IgM levels would eventually drop, or the local IgM levels in the plaque were considerably reduced. The authors should try running qPCR or a Western on aortic samples to determine local IgM levels.

8. Although stated in the discussion, it was not evident anywhere in the manuscript that high fat feeding profoundly alters the protective phenotype of ILC2 cells. This would need to be dramatically bolstered. Furthermore, the discussion was particularly weak and did not adequately discuss the biological relevance of these data and how these data add to the body of current knowledge and could lead to potential treatments for patients.

9. It is unclear why the authors spend time on evaluation of FALCs, both in histology and flow cytometry in Fig. 1C-F and S1A, S1F. FALCs were deemed an area low in ILC2 cell number, so it is unclear why data relating to FALCs shown or how it is relevant.

Minor:

1. The authors state in the abstract that ILC2s are present in the aortic adventitia - but this was only demonstrated in aged ApoE^{-/-} mice by histology (1 cell identified). The authors should also separate the adventitia and intima from the aorta at different levels and show the cellular population by flow cytometry.

2. There is an additional paper that should be cited and discussed regarding ILC2 involvement in atherosclerosis. Perry et al., 2013 found that reducing natural helper cell numbers (ILCs) through genetic ablation of Id3 resulted in reduced IL-5 levels and increased atherosclerosis.

3. It would be helpful to see flow plots of WT or young ApoE^{-/-} mice in addition to the plots shown in S1E and Fig. 1A to evaluate the reduction or absence in the ILC2 cells noted.

4. The authors state there was no difference in plasma lipid levels with IL-2-IL-2R treatment, in contrast to the Engelbertson paper. Nonetheless, the Engelbertson paper measured VLDL, LDL, and HDL, and only found a difference in VLDL (they did not report on total lipid content). Did the current authors also evaluate VLDL, LDL, and HDL separately to make their conclusions?

5. Rather than stating a "profound impact of splenectomy" on page 3, would the authors please identify whether this is a negative or positive impact?

Reviewers' comments:

Reviewer #1 (expert in immunology)

Remarks to the Author:

In the manuscript by Newland et al, the authors show that ILC2 are present in aortic adventitia, para-aortic adipose tissue and lymph nodes and that genetic ablation of ILC2 in *Ldlr*^{-/-} mice accelerated the development of atherosclerosis. This study is an extension of previous findings from other groups reporting the presence of ILC2 in aorta and their role in the development of atherosclerosis (Mantani et al, 2015; Engelbersten et al 2015). In comparison to these studies, the authors use a more selective approach of ILC2 depletion and show that accelerated development of atherosclerosis in *Ldlr*^{-/-} mice could be prevented by reconstitution with wild type but not *Il5*^{-/-} or *Il13*^{-/-} ILC2.

Response: Please, note that the 2 studies cited above by the reviewer did not provide any solid or direct evidence for a role of ILC2 in atherosclerosis. Mantani et al injected mice with IL-25 and observed some reduction of atherosclerosis (in the thoracic aorta but not at the level of the aortic root) concomitantly with increased expansion of ILC2. There was absolutely no direct demonstration that the small site-specific changes of atherosclerosis were due to ILC2 expansion. The same comment applies to the study by Engelbersten et al. who treated immunocompromised T and B cell deficient mice with IL2 and reported a reduction of atherosclerosis concomitantly with ILC2 expansion. No direct evidence was provided that the expanded ILC2 were responsible for the decrease of lesion size. Moreover, plasma cholesterol levels were significantly reduced after IL2 treatment, which may explain, at least in part, the reduction of atherosclerosis. We also report numerous side effects of IL-2 administration in immunocompromised mice. For all those reasons, such experiments cannot be relied on to determine the role of ILC2 in atherosclerosis (or in other disease states). This is discussed in our manuscript: "Previous studies suggested a potential role for ILC2 in the modulation of atherosclerosis^{23,26}. However, those studies used immune-compromised animals and relied on non-physiological exogenous and chronic administration of cytokines (i.e., IL-2 and IL-25) that are not specific for the ILC2 population, and that can promote ILC2-independent immune responses. Moreover, those studies failed to provide any direct evidence of the involvement of ILC2 in atherosclerosis and were confounded by profound alterations of hepatic and lipid metabolism²⁶ following chronic exogenous cytokine administration. We also found that chronic administration of IL-2/anti-IL2 complexes in immune-compromised animals reduced atherogenesis but in agreement with previous findings, the effect was associated with several adverse side effects".

Another point is that expansion of ILC2 by exogenous administration of cytokines does not tell anything about the role of the endogenous ILC2 population and its contribution to atherosclerosis.

None of the previous studies looked at the impact of disease progression or HFD on the phenotype of ILC2.

Thus, our work (now extensively revised) provides original data on ILC2 phenotype in different locations (spleen, lymph nodes, aorta, para-aortic lymph nodes, para-

aortic adipose tissue, etc.) during atherosclerotic disease progression on chow diet and in response to high fat feeding, and identifies for the first time a critical role for naturally-developing ILC2 (and more particularly ILC2-derived IL-13) in the modulation of aortic inflammation and atherosclerotic lesion development.

Reviewer: I have the following remarks:

In Figure 1A and relative text, the authors conclude that "the ILC2 population in para-aortic fat differed significantly from that of peri-gonadal fat, and comprised a substantially lower proportion of nILC2... favouring a predominance of iILC2 over nILC2". iILC2 cells are apparently not present in peripheral tissues in the steady state but can be elicited at many sites by helminth infection or IL-25 treatment (Huang and Paul, *Int Imm* 2016), while ST2 and KLRG1 expression differ *ex vivo* in different tissues. Thus, I would avoid the term iILC2 (unless phenotype analysis is performed after *in vivo* IL-25 treatment).

Response: We have altered our description of these cells to reflect this, based on the surface expression of ST2.

Reviewer: Moreover, similar to previous publications (Mantani et al, 2015; Engelbersten et al 2015), in this study only a couple of ILC2 markers have been evaluated. Further phenotypic and functional characterization of ILC2 in the paraaortic LN and AT compared to mesenteric LN and lung (or small intestine), where they represent a major ILC population, should be accomplished *ex vivo*. In addition to ICOS, KLRG1 and ST2, *ex vivo* staining of GATA3, Sca-1, CD25, IL17BR, CD127, c-kit, CD90 and cytokine expression (after *ex vivo* stimulation) should be performed.

Response: We regularly include CD25 and CD127 and KLRG1, which are the more consistent of surface markers for ILC2 in our panel, examples of which are included in supplemental figure 1E. The text has been amended to reflect this. In other parts of the manuscript (chow versus HFD), ILC2 were sorted from spleen, aorta, para-aortic adipose tissue, GWAT, and gene expression analysis performed after cell sorting.

Reviewer: In the text, page 6 the authors state: "ILC2, which were not found in aortic adventitia of WT or very young Apoe^{-/-} mice (7 weeks old, not shown), increasingly accumulate a mixture of nILC2 (KLRG1⁺ST2⁺) and iILC2 (KLRG1^{high}ST2⁻ or low) in the adventitia of Apoe^{-/-} mice as disease progressed with age (Figure 1B), and were found in ATLO of old Apoe^{-/-} mice with advanced atherosclerosis (Figures S1D and S1E)".

No data supporting this conclusion are shown.

Response: We have now amended this figure to show the change in population dynamics of Lin⁻ ICOS⁺ CD25⁺ CD127⁺ KLRG1⁺ (shown as %ILC2 in the figure) during aging of Apoe^{-/-} mice as Figure 1C. Further we describe the change in ST2 MFI in these different compartments.

Reviewer: The authors need to show a quantitative and qualitative comparison of ILC2 frequencies, absolute numbers, phenotype and *ex vivo* cytokine production in

paraortic LN and AT (compared to mLN and lung, or SI as control) in Apoe^{-/-} and WT mice at different time points during aging (or disease progression in Apoe^{-/-} mice). Quantitative analysis should be performed both by Flow cytometry and histology.

Response: Please, note that this is an atherosclerosis study. In the context of atherosclerosis we do not feel that studying Lung or small intestine (SI) would be necessary to the manuscript. However, in Figure 1, we have compared the phenotype of ILC2 in para-aortic adipose tissue to that of peri-gonadal adipose tissue (the latter being well known to accumulate a large population of ILC2). Moreover, we include data on ILC2 frequencies and phenotype in aortic adventitia proper and para-aortic lymph nodes, compared to mesenteric lymph nodes. We have amended supplemental figure 2 to include absolute numbers that can be recovered from each of these tissues (Figure S2A). We feel that given the limited number of ILC2 in the peri aortic environment, quantification by histology would not result in an accurate description of the ILC2 population in these tissues.

The proper effect of aging on ILC2 phenotype throughout the body is not addressed in this manuscript. However, we analysed the effect of HFD on ILC2 phenotype in the aorta, spleen, MLN and GWAT. Data is presented in revised Figure 2.

Reviewer: Moreover, the authors show that "FALCs develop in the para-aortic WAT of old WT and Apoe^{-/-} mice", but "accumulated only a few ILC2". A quantification of ILC2 at this site and a comparison between numbers in old WT and Apoe^{-/-} mice would be again very helpful.

Response: As described above, histological quantification may not be accurate enough to describe this population of cells in FALC, however we have now included flow cytometry data comparing young (7 weeks) and Aged (80+ weeks) Apoe^{-/-} mice vs WT PaAT resident ILC2 (Figure S2B and C).

Overall, the data indicate that ageing of Apoe^{-/-} mice on chow diet does not alter the number of PaAT ILC2. However, the number of ILC2 increases in aortic adventitia with age.

Reviewer: Page 9: "To ensure the bone marrow graft was effective and very few endogenous ILC2 remained, IL-33 was given 24h prior to organ harvest": the reason why the authors inject IL-33 in this experimental setting is not very clear to me. The authors should show that recipient ILC2 are indeed efficiently depleted by irradiation in all organs. Although BM ILC2 seem efficiently depleted in the BM, in MLN there is only a 3 fold reduction of ILC2 (Fig 3B). What about ILC2 in paraortic LN and other organs? If recipient ILC2 remained, I cannot see how IL-33 could be of help as it would also activate and expand those ones.

Response: There are two possible variables during a bone marrow transplant, which may affect the outcome of chimeric knockout models such as this. 1) the irradiation suitably eradicates the endogenous population of ILC2 and 2) the ILC2 can replenish all (lymphatic and tissue specific) compartments. Figure 4 has now been modified to demonstrate this more clearly. First, we have included data using Thy1.1/Thy1.2 transgenic mice to show that after irradiation and bone marrow reconstitution the

ILC2 present in both the lymphatic compartment as well as tissue resident cells are effectively replaced with donor cells.

IL-33 was administered 24h prior to harvest and thus the expansion of ILC2 at this time point would not affect the outcome of disease progression. However, it would demonstrate that endogenous ILC2 have been 'eradicated' in recipients of ILC2^{KO} bone marrow. We demonstrate that ILC2 are effectively depleted. This is shown using flow cytometry of ILC2 and detection of suppressed IL-5 and IL-13 serum response (new data in now Figure 4). We then go on to show that by RT-PCR the cytokine environment in both the aorta and the PaAT express less ILC2 "signature" cytokines IL-5 and IL-13 after transplant with ILC2^{KO} bone marrow (new data Figure 4D). Again, we agree that ILC2 depletion may not be complete in our setting. However, it is substantial and largely enough to impact aortic inflammation and lesions development.

Reviewer: Figure 4: the authors state that "ILC2 favor alternative activation of macrophages through production of IL-5 and IL-13". Most of these data have been generated in vitro, by stimulating splenic or peritoneal macrophages with ILC2 conditioned media. It would be important to validate these data in vivo by analyzing macrophage phenotype in paraortic LN and AT of old Apoe^{-/-} mice and WT mice in the absence of ILC2 or of ILC2 lacking IL-5 or IL-13.

Response: We agree that more in-depth analysis was needed. Therefore these data have been removed to supplemental figure 5 and an entirely new figure addressing these points has been created (figure 5) and heavily modified the final figure (now figure 6). We analysed macrophage phenotype locally in the aorta using immunohistochemistry and flow cytometry. Here we show changes in plaque and specifically plaque macrophage phenotype when ILC2 or ILC2 derived IL-13 is absent.

Minor comments:

Reviewer: It is not clear from Figure legends or Mat and Methods whether CD45 staining is performed or not. Gating on CD45⁺ Lin⁻ cells from tissue would be important to better define ILCs.

Response: In lymphoid tissues such as spleen and LN CD45 staining was not included. We use a comprehensive Lineage panel to exclude many cell types including Ter119 labelling on red blood cells. The vast majority of cells present in the initial lymphocyte gate are CD45⁺ and therefore it would give no additional information. However in tissues such as GWAT and Aortic AT where large numbers of contaminating cells would also be lineage negative CD45⁺ has been included.

Reviewer: Legend to Figure 1G mentions a Figure 1G which is not there (and is also not commented in results)

Response: Our apologies we have modified the figure legends to accommodate this and other changes

Reviewer: Figure 2A-B: the authors define ILC2 by using different markers in different compartments. Gata3 staining could be used.

Figure 2C: cytokine expression should be evaluated ex vivo, not after in vitro culture

Response: Including GATA3 in flow cytometry panels may not help when identifying ILC2, GATA3 is quite permissive and expressed in ILC3 as well as ILC2 (ref Serafini et al 2014 Zong et al 2016). We do however now include RT-PCR of IL-5, IL-13 and GATA3 in ex vivo sorted ILC2 from spleen, GWAT and Aorta after 8 weeks of high fat diet (new Figure 2D). Here it is interesting to show that while the expression of these cytokines is suppressed in the spleen of HFD mice (mirroring the ex vivo cytokine analysis) the expression is increased in the aorta, indicative of an increase in ILC2 activation during aortic inflammation.

Reviewer: Figure 2D: percentages and absolute numbers of ILC2 after IL-2 complex treatment should be quantified in the paraortic LN/adventitia and not only in BM and spleen

Response: We have included data from the MLN to supplement the expansion of ILC2 during IL-2 treatment. Our observations by immunofluorescence could detect clusters of ILC2 in the aortic adventitia of IL-2 treated mice, which were absent (zero ILC2 cluster seen by immunohistochemistry) in the PBS cohort. Given that we (and others) have observed significant side effects of IL2 administration in immunocompromised mice, which may affect disease progression (fibrosis in a number of organs for example) and we conclude that it is not necessarily suitable to expand ILC2 in this way in RAG mice we do not think it is required to perform and show a complete survey of ILC2 here.

Reviewer: Legend to Figure 4 does not have C and D in the description.

Response: We apologise for omitting the description of previous Fig4 C and D and the text has been amended to include them.

Reviewer #2 (expert in atherosclerosis and immunology)

Remarks to the Author:

The authors show that depletion of ILC2 cells accelerates atherosclerosis, which can be rescued by adoptive transfer of wild-type, but not IL-13 or IL-5 deficient ILC2 cells. This is not entirely unexpected, because a 2015 ATVB paper showed similar results. There is another paper (ref 23) that also implicates ILC2 cells in atherosclerosis, but is not cited for this aspect.

Response: We really do not think that it is possible to argue that the 2015 ATVB paper 'showed similar results'. The 2 studies cited above by the reviewer did not provide any solid or direct evidence for a role of ILC2 in atherosclerosis. Mantani et al injected mice with IL-25 and observed some reduction of atherosclerosis (in the thoracic aorta but not at the level of the aortic root) concomitantly with increased expansion of ILC2. There was absolutely no direct demonstration that the small site-specific changes of atherosclerosis were due to ILC2 expansion. The same comment applies to the study by Engelbersten et al. who treated immunocompromised T and B cell deficient mice with IL2 and reported a reduction of atherosclerosis concomitantly with ILC2 expansion. No direct evidence was provided that the expanded ILC2 were responsible for the decrease of lesion size. Moreover, plasma cholesterol levels were

significantly reduced after IL2 treatment, which may explain, at least in part, the reduction of atherosclerosis. We also report numerous side effects of IL-2 administration in immunocompromised mice. For all those reasons, such experiments cannot be relied on to determine the role of ILC2 in atherosclerosis (or in other disease states). This is discussed in our manuscript: “Previous studies suggested a potential role for ILC2 in the modulation of atherosclerosis^{23,26}. However, those studies used immune-compromised animals and relied on non-physiological exogenous and chronic administration of cytokines (i.e., IL-2 and IL-25) that are not specific for the ILC2 population, and that can promote ILC2-independent immune responses. Moreover, those studies failed to provide any direct evidence of the involvement of ILC2 in atherosclerosis and were confounded by profound alterations of hepatic and lipid metabolism²⁶ following chronic exogenous cytokine administration. We also found that chronic administration of IL-2/anti-IL2 complexes in immune-compromised animals reduced atherogenesis but in agreement with previous findings, the effect was associated with several adverse side effects”.

Another point is that expansion of ILC2 by exogenous administration of cytokines does not tell anything about the role of the endogenous ILC2 population and its contribution to atherosclerosis.

None of the previous studies looked at the impact of disease progression or HFD on the phenotype of ILC2.

Thus, our work (now extensively revised) provides original data on ILC2 phenotype in different locations (spleen, lymph nodes, aorta, para-aortic lymph nodes, para-aortic adipose tissue, etc.) during atherosclerotic disease progression on chow diet and in response to high fat feeding, and identifies for the first time a critical role for naturally-developing ILC2 (and more particularly ILC2-derived IL-13) in the modulation of aortic inflammation and atherosclerotic lesion development.

1. Reviewer: It is surprising that IL5 and IL13 are both critically required in ILC2 cells as suggested in figure 4. No explanation is given for this.

Response: We have been able to identify the mechanism for IL13 mediated effect though the alternative activation of macrophages (see revised Figures 5 and 6) however we are not yet convinced that IL5 functions in a similar manner. For example, ILC2-derived IL5 had no significant effect on macrophage phenotype. It is particularly interesting that the impact of ILC2 selective IL-5 deficiency appears to be site specific (eg a difference was observed in aortic arch but no significant difference was present in aortic sinus). Previous Figure 3 (now Figure 4) has since been expanded to show that in the absence of ILC2 (ILC2ko BMT) there is a decrease of IL-13 and IL-5 expression in both the aorta and peri aortic AT as detected by RT PCR. It is thus sensible to observe the effect of both these cytokines in the context of ILC2 secretion. We have amended the figures and text to account for these changes.

2. Reviewer: The criticism of ref 26 is not fair. Similar to reference 26, the authors also use ILC2 cells expanded in Rag KO mice.

Response: The criticism is fair because it is accurate. In fact, we observed quite similar side effects as previous ef26 and criticize our own data by stating that there are considerable problems inherent to this model such as the increased fibrosis after long term IL-2 treatment. This is a process that may affect atherosclerosis progression and we therefore state that it is preferential to allow the complete immune system to function in the absence of ILC2. Again, we believe this criticism is fair and we explain that we cannot rely solely on IL-2 (or IL-25) mediated expansion or antibody mediated depletion of ILC2 (which is not specific for ILC2) to address the role of ILC2 in atherosclerosis.

3. Reviewer: The authors looked for ILC2 cells in the para-aortic lymph nodes and the para-aortic adipose tissue, but not in the aorta proper. Why?

Response: We assume that the reviewer is referring to the first figure with this comment (although this is not clear). Here we clearly state that ILC2 are detected in the aortic adventitia as in Fig 1B.

4. Reviewer: The image pair shown in figure 3C does not appear to be representative of the data.

Response: We thank reviewer 2 for his/her close observations of our representative data. We feel it is difficult to state this when the data we present is a function of plaque area by depth through the aortic sinus, however we are happy to include further sections for their satisfaction.

5. Reviewer: ATLOs are really not observed in mice of the age studied here. They usually appear after 1 year of age.

Response: The reviewer is not clear which figure this refers to. We are assuming they are referring to figure one. Fig1 A and B describe the presence of ILC2 in mice at 24 weeks of age. We then describe the presence of ATLO specifically in mice 70-80 weeks old. We have now added more data on the distribution of ILC2 in young and aged Apoe^{-/-} mice on chow diet.

6. Reviewer: In the introduction, the authors acknowledge that the role of Th2 cells in atherosclerosis is unknown (controversial), but in the methods they claim that type II cytokines are atheroprotective. This is highly controversial. For example, the Th2 disease asthma is known to exacerbate atherosclerosis.

Response: We state in the introduction that Th1 cells can exacerbate atherosclerosis but the effects of Th2 and Th17 cells is more complex and they may either inhibit or exacerbate the disease which we feel is not a controversial statement. We then go on to show examples in the literature where type II cytokines, namely IL-5 and IL-13, have a direct protective effect on plaque progression. We would be happy to change our statement according to any specific suggestion from the reviewer. We have now altered the sentence to indicate that 'some' type II cytokines are atheroprotective, and listed IL-5 and IL-13 as main examples.

7. Reviewer: The mouse model is not introduced. Why would CD127-Cre driven KO of RORa lead to specific depletion of ILC2 cells?

Response: Reference 36 of the manuscript describes and validates this mouse model of ILC2 specific deficiency.

8. Reviewer: Figure 3: the lesion size is small. 9 weeks HFD is probably too short for Ldlr^{-/-} mice.

Response: Mean lesion size in previous Figure 3 (now Figure 4) varies between 200 000 μm^2 and more than 300 000 μm^2 . This is an advanced stage of lesion development. Mean lesion size in previous Figure 4 (now Figure 6) varies between 300 000 μm^2 and more than 450 000 μm^2 . This is again an advanced stage of lesion development.

9. Reviewer: What are "normal conditions" for expanding ILC2 cells? The data shown in figure 4A seem to be from a highly contrived in vitro system.

Response: We recognise the weakness of the M1/M2 in vitro work. We have now added an additional figure including ex vivo scoring data (immunohistochemistry and flow cytometry on aorta) showing a M2 deficiency when ILC2 are absent as well as when ILC2 are present but unable to produce IL-13 (Figure 5 and Figure 6). The text has been amended to clearly reflect this change.

10. Reviewer: The diets are not sufficiently defined: "normal Chow (manufacturer) or Western High 330 Fat Diet (Dietex)". Western diet is not really suitable in Ldlr^{-/-} mice unless cholesterol is also high.

Response: The normal chow and Western diets have been more adequately defined in the new text. Western diet is highly cited for atherosclerosis studies in LDLR deficient mice. It contains 0.25% cholesterol on top of the high fat content.

11. Reviewer: Statistical concern: It appears that multiple comparisons were not corrected for.

Response: There is only one experiment (now Figure 6A) where performing multiple comparisons was appropriate. This has been done using Kruskal Wallis test, which showed significance. Then, as in all other cases, the non parametric Mann Whitney U test has been used to compare discrete groups unless otherwise stated in the figure legends.

Reviewer #3 (expert in atherosclerosis and immunology)
Remarks to the Author:

In the paper "ILC2 control the development of atherosclerosis in mice" Newland et al utilize a number of different approaches to demonstrate that ILC2 reside in various regions of the aorta, that they can produce IL-13 and IL-5, that they restrict development of atherosclerosis, and that they presumably do so by influencing

macrophage polarization. There have been at least two papers recently published that specifically investigated the role of ILC2 in atherosclerosis that arrived at the same conclusion (Engelbertson et al 2015; Mantani et al 2015). Other papers have implied this, while a protective role for IL-5 is known. Much of the data shown in the manuscript are therefore a repeat of those findings.

Response: We really do not think that the previous knowledge on the protective role for IL-5 and IL-13 in atherosclerosis truly limits the novelty of our paper. We also do not think that it is possible to argue that the 2 studies cited above have provided direct evidence for a role of ILC2 in atherosclerosis. Mantani et al injected mice with IL-25 and observed some reduction of atherosclerosis (in the thoracic aorta but not at the level of the aortic root) concomitantly with increased expansion of ILC2. There was absolutely no direct demonstration that the small site-specific changes of atherosclerosis were due to ILC2 expansion. The same comment applies to the study by Engelbersten et al. who treated immunocompromised T and B cell deficient mice with IL2 and reported a reduction of atherosclerosis concomitantly with ILC2 expansion. No direct evidence was provided that the expanded ILC2 were responsible for the decrease of lesion size. Moreover, plasma cholesterol levels were significantly reduced after IL2 treatment, which may explain, at least in part, the reduction of atherosclerosis. We also report numerous side effects of IL-2 administration in immunocompromised mice. For all those reasons, such experiments cannot be relied on to determine the role of ILC2 in atherosclerosis (or in other disease states). This is discussed in our manuscript: “Previous studies suggested a potential role for ILC2 in the modulation of atherosclerosis^{23,26}. However, those studies used immune-compromised animals and relied on non-physiological exogenous and chronic administration of cytokines (i.e., IL-2 and IL-25) that are not specific for the ILC2 population, and that can promote ILC2-independent immune responses. Moreover, those studies failed to provide any direct evidence of the involvement of ILC2 in atherosclerosis and were confounded by profound alterations of hepatic and lipid metabolism²⁶ following chronic exogenous cytokine administration. We also found that chronic administration of IL-2/anti-IL2 complexes in immune-compromised animals reduced atherogenesis but in agreement with previous findings, the effect was associated with several adverse side effects”.

Another point is that expansion of ILC2 by exogenous administration of cytokines does not tell anything about the role of the endogenous ILC2 population and its contribution to atherosclerosis.

None of the previous studies looked at the impact of disease progression or HFD on the phenotype of ILC2.

Thus, our work (now extensively revised) provides original data on ILC2 phenotype in different locations (spleen, lymph nodes, aorta, para-aortic lymph nodes, para-aortic adipose tissue, etc.) during atherosclerotic disease progression on chow diet and in response to high fat feeding, and identifies for the first time a critical role for naturally-developing ILC2 (and more particularly ILC2-derived IL-13) in the modulation of aortic inflammation and atherosclerotic lesion development.

Reviewer: That said, the authors do use a genetic model that specifically depletes ILC2, but the analysis using this model is somewhat thin and rudimentary.

Response: We have used a genetic model that specifically depletes ILC2 and harnessed the model to reconstitute with WT, IL5-deficient or IL13-deficient ILC2. We do not understand why this is considered by the reviewer to be thin and rudimentary. We do not know of any better approach to do it.

Reviewer: Moreover, the mechanistic analysis, that is, the suggestion that ILC2 protect against atherosclerosis by polarizing macrophages is very weak because the authors use only splenic and peritoneal macrophages and look at a few markers in vitro. As such, we have no idea whether ILC2 protect against atherosclerosis locally by acting on macrophages in vivo or whether that protection occurs elsewhere.

Response: As stated for Reviewer 2, we agree that the section describing macrophage polarisation is weak and have thus provided extensive ex vivo analysis of macrophage phenotype in the plaque and plaque phenotype. We now show (by flow, IHC and RT PCR) that in the absence of ILC2 (now figure 4) as well as when ILC2 are deficient for IL-13 production (new figures 5 and 6) not only is plaque size increased but the M1/M2 polarisation is skewed towards M1.

Reviewer: As a very minor population, we also have no idea how much of a IL-5 and IL-13 source ILC2 are in atherosclerosis.

Response: We now show in new Figure 4 the impact of ILC2 deletion on IL5 and IL13 locally and systemically. We have also directly addressed the contribution of ILC2-derived IL-5 or IL-13 to atherosclerosis as shown in the mixed bone marrow chimeras (new figure 6).

Reviewer: Ultimately, the paper presents an incremental advance and fails to deliver a solid mechanistic analysis.

Response: Our work (now extensively revised) provides original data on ILC2 phenotype in different locations (spleen, lymph nodes, aorta, para-aortic lymph nodes, para-aortic adipose tissue, etc.) during atherosclerotic disease progression on chow diet and in response to high fat feeding, and identifies for the first time a critical role for naturally-developing ILC2 (and more particularly ILC2-derived IL-13) in the modulation of aortic inflammation and atherosclerotic lesion development. We hope the reviewer will agree now that the paper provides significant advance over current knowledge.

Specific comments

Major:

1. Reviewer: There have been two previous publications evaluating the role of ILC2 cells in atherosclerosis: Engelbertsen et al., 2015 and Mantani et al., 2015. The current manuscript spends a lot of time justifying the need for the current studies because the previous publications were flawed. Nonetheless, this manuscript actually approaches the question of ILC2 cells in atherosclerosis the exact same way, with the exception of an additional genetic model in Figure 3 and Figure 4. It is important to note that Figures 1 and 2 conform almost exactly with the data shown in the Engelbertsen and

Mantani papers, significantly reducing the novelty of the current manuscript.

Response: Please, see our response above. The reviewer is correct in their summation that we initially used the same approach as previously published. However, we found significant deficiencies in this approach and designed a model system, which avoided these deficiencies.

The previous studies did not provide details about ILC2 phenotype in different aortic and para-aortic compartments, during disease progression or directly comparing chow versus high fat diet.

The use of the ILC2 specific knockout model is not incremental. It is a major step forward and allows for the first time to draw solid conclusions about the role of ILC2 in atherosclerosis. Moreover, the genetic model is harnessed to reconstitute with WT, IL-5 deficient and IL-13 deficient ILC2, which provides invaluable and unprecedented information on their specific roles in atherosclerosis. We also show via that the phenotype of plaque macrophages is altered in the absence of ILC2 or ILC2-derived IL-13, skewed towards the M1 phenotype.

2. Reviewer: It was unclear which population of ILC2 cells the authors identified as the most important during the progression of atherosclerosis. The authors note that very few ILC2 cells are present in the para-aortic lymph nodes (LN) and para-aortic adipose tissue (AT), but show a considerable number of cells in the abdominal and thoracic aorta. The importance of each population is unclear, however, because despite the dominant ILC2 population in the aorta, the entire manuscript focuses on ILC2 cells in the mesenteric LN and para-aortic LN.

Response: We now provide extensive data on ILC2 in the aorta and para-aortic adipose tissue in different atherosclerosis setting, and in response to HFD. Moreover, we document the impact of ILC2 deficiency on plaque macrophage phenotype. We agree, we as yet do not understand which population of ILC2 are most important for decreasing lesion size. We do show however that ILC2 have a niche in the tissues adjacent to the aorta and when they are absent from this niche there is an increase in atherosclerosis severity. We look forward to investigating this further.

3. Reviewer: It is unclear how ILC2 cells increase in the aorta throughout the progression of atherosclerosis, but are reduced by high fat diet.

Response: We apologize for any misunderstanding. The increase of ILC2 in the aorta (adventitia) is seen in Apoe^{-/-} mice on chow diet between 7 weeks and 24 weeks of age (new Figure 1C). The decrease of ILC2 is seen in spleen, mesenteric and para-aortic lymph nodes in Ldlr^{-/-} mice after HFD (Figure 2B). We have now also cell sorted ILC2 from aortas of Ldlr^{-/-} mice on HFD and show in Figure 2D that they behave differently from splenic ILC2. The new data is discussed in the manuscript.

4. Reviewer: For the analysis in Fig. 2C, the authors state that after expansion of ILC2 cells ex vivo, less IL-13 and IL-5 were produced by ILC2 cells generated from HFD-treated mice. Fewer ILC2 cells were collected from these mice as well. Were equal numbers of cells plated ex vivo and did the authors confirm equal expansion of both ILC2 cell types? If there were fewer ILC2 cells from the HFD-treated mice, or the cells were less responsive to IL-7/IL-33, then the lower levels of IL-13 and IL-5

would result from the fact that there were fewer cells, not because they were qualitatively different in cytokine expression.

Response: Equal numbers of cells were used for each condition. The methods section has now been altered to state that “ex vivo sorted ILC2 cells were plated at a density of 5×10^4 cells per well before restimulation with IL-7 and IL33”.

We have now also cell sorted ILC2 and directly performed gene expression analysis without any culture or restimulation step. The data is presented in Figure 2D and confirm the above-mentioned data.

5. Reviewer: Overall the ex vivo data seem particularly weak. These data do not support a strong case for actions of IL-5 and IL-13 on macrophages as the driver behind reduced atherosclerosis with more ILC2 cells. Indeed, iNOS and TNF α , two factors strongly implicated in atherogenesis, remained unchanged with ILC2 treatment of macrophages. Moreover, these macrophage-associated effects were never evaluated in vivo. If the effects were indeed entirely related to changes in arginase-1 and fizz1 gene expression in macrophages, then the authors should show commensurate changes in plaque stability. Also, why would changes in arginase-1 and fizz-1 in macrophages affect how many macrophages accumulate within the plaque? Each of these factors needs to be addressed before macrophages can be implicated in the mechanism of reduced atherosclerosis in this model.

Response: As mentioned above the macrophage and plaque phenotypes have now been extensively described. The data strongly argue for a major role of ILC2 and ILC2-derived IL-13 in modulating plaque macrophage phenotype (new Figure 4C, 4D, 5B, 5C, 6C, 6D, and supplement).

6. Reviewer: It is also unclear why the authors chose to do in vivo experiments with IL-5. IL-5 showed no great potential ex vivo and looked like it may have actually increased iNOS expression in peritoneal macrophages.

Response: If we had not done it, the reviewer would certainly have requested it. IL-5 is one of the major cytokines secreted by ILC2 and may have an effect distinct from the alternative activation driven by IL-13. The RT PCR data now included in Figure 4 shows a decrease in expression of both IL-5 and IL-13 after ILC2 deletion. It is prudent to analyse the effect of ILC2 derived IL-5 for a better understanding of this cell type.

7. Reviewer: The conclusion to exclude a role for IgM seems a bit premature. There were fewer B1 cells in several tissues investigated, and it is possible IgM levels would eventually drop, or the local IgM levels in the plaque were considerably reduced. The authors should try running qPCR or a Western on aortic samples to determine local IgM levels.

Response: We have now included histology data showing no difference between IgM deposition in ILC2^{WT} or ILC2^{KO} plaques.

8. Reviewer: Although stated in the discussion, it was not evident anywhere in the manuscript that high fat feeding profoundly alters the protective phenotype of ILC2 cells. This would need to be dramatically bolstered. Furthermore, the discussion was

particularly weak and did not adequately discuss the biological relevance of these data and how these data add to the body of current knowledge and could lead to potential treatments for patients.

Response: New Figure 2 is entirely dedicated to the study of ILC2 phenotype in response to HFD. The new data is discussed within the Results and Discussion section. The latter is short given restriction on word limit.

9. Reviewer: It is unclear why the authors spend time on evaluation of FALCs, both in histology and flow cytometry in Fig. 1C-F and S1A, S1F. FALCs were deemed an area low in ILC2 cell number, so it is unclear why data relating to FALCs shown or how it is relevant.

Response: They may be low in number but are a niche the ILC2 are present in adjacent to the aorta and thus give the ILC2 a localised sensing environment that they can expand and respond from. Further although FALC have been described in a number of tissues this is the first time they have been described in the aortic context.

Minor:

1. Reviewer: The authors state in the abstract that ILC2s are present in the aortic adventitia - but this was only demonstrated in aged ApoE^{-/-} mice by histology (1 cell identified). The authors should also separate the adventitia and intima from the aorta at different levels and show the cellular population by flow cytometry.

Response: ILC2 are neither present in the intima nor in the media. The flow cytometry data presented in fig1a, 1b and 1c show the presence of ILC2 in the aorta (aortic adventitia) by flow cytometry. In Figure 2D, we cell-sorted ILC2 from the aorta and analysed changes in gene expression comparing mice on chow versus HFD. In Figure 3b we show that ILC2 are expanded via IL-2 in the adventitia. We have amended the text to make this more apparent.

2. Reviewer: There is an additional paper that should be cited and discussed regarding ILC2 involvement in atherosclerosis. Perry et al., 2013 found that reducing natural helper cell numbers (ILCs) through genetic ablation of Id3 resulted in reduced IL-5 levels and increased atherosclerosis.

Response: Not a problem we have put it in context.

However, please note that in Perry HM et al., ATVB 2013: Id3^{-/-} mice show increased atherosclerosis. The mice have normal (not reduced) ILC2 levels. When mice are stimulated with exogenous IL-33, their natural helper cells (ILC2) produced less IL5 than Id3^{+/+} mice. However, this is a total Id3 knockout and many other immune and vascular cells are altered by the absence of Id3 (as shown before by many other groups). No experiment was presented to show that the reduction of atherosclerosis was due to the small alteration of IL5 production by ILC2 (observed only after exogenous IL33 stimulation).

3. Reviewer: it would be helpful to see flow plots of WT or young ApoE^{-/-} mice in addition to the plots shown in S1E and Fig. 1A to evaluate the reduction or absence in the ILC2 cells noted.

The requested plots have now been included in figure S2.

4. Reviewer: The authors state there was no difference in plasma lipid levels with IL-2-IL-2R treatment, in contrast to the Engelbertson paper. Nonetheless, the Engelbertson paper measured VLDL, LDL, and HDL, and only found a difference in VLDL (they did not report on total lipid content). Did the current authors also evaluate VLDL, LDL, and HDL separately to make their conclusions?

Response: We invite the reviewer to have a look at Figure 4D of the Engelbertson paper. There was a significant reduction in total plasma cholesterol levels.

5. Reviewer: Rather than stating a "profound impact of splenectomy" on page 3, would the authors please identify whether this is a negative or positive impact?

Response: We replaced 'splenectomy' by 'spleen-dependent responses'. It would be out of the scope of the manuscript to detail the positive and negative impact of splenic responses. The reader is provided with appropriate references.

Reviewers' comments:

Reviewer #3 (Remarks to the Author):

MAJOR

1. While the authors have now provided some flow cytometric and qPCR data on Arginase and iNOS in the plaque macrophages, it is still unclear as to whether this alteration in phenotype is causing a change in atherosclerosis. Particularly because there was no change in collagen deposition or smooth muscle cell concentration between the ILC2WT and ILC2KO groups. Moreover, this alternation in phenotype does not explain why a higher number of macrophages would be found in the plaque. Are these cells proliferating? Are a higher number of monocytes immigrating to the plaque? This aspect of the manuscript is still rather weak. Particularly because the authors do not investigate both Arginase and iNOS in the IL-13 reconstitution study. At least a discussion on this would be helpful.

2. It is also unclear why collagen deposition is unchanged between ILC2WT and ILC2KO groups, but is lower in ILC2KO-IL-13- mice compared to ILC2KO-IL-13+ mice. Please comment.

3. Moreover, the authors now show a drop in ILC2 levels in the periphery, but no change in the local aortic populations. How, then, do the authors explain the drop in IL-13 and IL-5 levels in the aorta?

4. It is unclear why the authors compare ILC2KO-IL-5 groups with ILC2WT groups, but separate ILC2KO-IL-13 mice into a separate figure. Is there a reason why ILC2KO-IL-13 mice were not directly compared with ILC2WT mice?

5. The authors show a significant drop in atherosclerosis in ILC2KO-IL-5+ mice, but IL-5 had little to no effect on macrophage phenotype. The authors do not provide a potential explanation as to why IL-5 can now rescue the ILC2KO phenotype.

MINOR

In Supplemental Figure 4, there is no D to which the authors reference and there is no lesion size noted for B.

Reviewer #4 (replacement for original Referee #1)

Remarks to the Author:

The authors should adequately respect the previous papers (refs. 23, 26, and 27 of the revised manuscript) and reviewers' comments. Nevertheless, employment of *Rora*^{<fl/fl>}*Il7ra*^{<Cre>} mice is an important step to prove the importance of ILC2 in the disease progression, which added our understanding of the role of ILC2 in atherosclerosis. It is preferable for authors, however, to state that because ILC2 are tissue-resident cells, whole body irradiation may not deplete ILC2 present in peripheral tissues and it is difficult to completely deplete ILC2 in *Rora*^{<fl/fl>}*Il7ra*^{<Cre>} mice.

I found several minor points that need to be clarified.

1) Page 6, lines 2 and 5; Fig. 1B should not be referred to because Figure 1B does not show the surface phenotypes.

2) Page 6, line8; Figure 1C does not contain data on spleen.

3) Figure S1C and S2A are not referred to in the text. Figure S2B and S2C are referred to before Figure S2D and S2E.

4) Page 8 and Figure 2; the authors showed the decreased expression of IL-5 and IL-13 by ILC2 in MLN and spleen but not by those in aorta. Although the authors studied the role of ILC2s in aorta is involved in the pathogenesis of atherosclerosis, why do the authors pick up the phenotype of ILC2s in MLN and spleen?

Reviewer #5 (replacement for original referee #2)

Remarks to the Author:

This revised manuscript examines the key role for ILC2 in atherosclerosis. The data are reasonably convincing if you consider that what is shown is a global, possibly systemic, role of ILCs in the disease. Apparently, they do this by secreting IL5 and IL13. The claimed measurement of ILCs in the aortic adventitia (Fig. 2B) while never showing the gating or plots therein confuses the issue of location as to where the ILCs are working.

1. In fact, I am disappointed by the gating of ILCs overall in the plots that are shown and the fact that the adventitial gating is not even shown might suggest that it is poor or unconvincing. This should be shown or removed from the manuscript, with the manuscript being adjusted to take a more global (systemic) view of their role.

2. Also, the M2 polarization aspect of the story is especially unconvincing. The antibody used to Arg1 is not well validated and in any case, a single marker (Arg1 and iNOS) have been used to mark M2 and M1 macrophages, respectively. I believe that this aspect of the manuscript should be removed. There is no direct evidence of the mechanistic link and the actual evidence for skewed polarization is weak.

Responses to Reviewers

Reviewer #3 (Remarks to the Author):

MAJOR

1. While the authors have now provided some flow cytometric and qPCR data on Arginase and iNOS in the plaque macrophages, it is still unclear as to whether this alteration in phenotype is causing a change in atherosclerosis. Particularly because there was no change in collagen deposition or smooth muscle cell concentration between the ILC2WT and ILC2KO groups. Moreover, this alteration in phenotype does not explain why a higher number of macrophages would be found in the plaque. Are these cells proliferating? Are a higher number of monocytes immigrating to the plaque? This aspect of the manuscript is still rather weak. Particularly because the authors do not investigate both Arginase and iNOS in the IL-13 reconstitution study. At least a discussion on this would be helpful.

Response: As requested, we have now acknowledged those limitations and discussed the potential mechanistic links between ILC2-derived type 2 cytokines, macrophage phenotype and changes in plaque size and composition. Please, see page 14: "IL-5-dependent atheroprotection was limited to the thoracic aorta and could not be attributed to changes in macrophage phenotype or B1-dependent natural IgM production. IL-13-dependent atheroprotection was associated with important changes in collagen deposition and macrophage phenotype, suggestive of alternative activation. However, the direct links between changes of macrophage phenotype and atheroprotection were not addressed".

2. It is also unclear why collagen deposition is unchanged between ILC2WT and ILC2KO groups, but is lower in ILC2KO-IL-13- mice compared to ILC2KO-IL-13+ mice. Please comment.

Response: Please, note that the absence of change in collagen deposition between ILC2WT and ILC2KO groups despite the larger plaque size in ILC2KO highly suggests impaired collagen deposition in the absence of ILC2. Indeed, the natural progression of atherosclerotic lesions is associated with increased collagen deposition, i.e., larger plaques accumulate more collagen. This is apparent when you compare collagen deposition in plaques of different size in the WT mice: 6% collagen for a plaque size of $\sim 200,000 \mu\text{m}^2$ in Fig S5D and Fig 4F versus 11% collagen for $\sim 300,000 \mu\text{m}^2$ in Fig 6D and 6B. However, collagen deposition is much lower (50% lower) in ILC2KO plaques when considering equivalent plaque size: only 5% collagen for $\sim 300,000 \mu\text{m}^2$ in Fig S5D and Fig 4F. If ILC2 deficiency had no effect on collagen deposition, the % collagen for a mean plaque size of $\sim 300,000 \mu\text{m}^2$ would have been equivalent to 11%, as in the WT. A summary of this is given in the manuscript on page 12: "It is usual for larger, more advanced plaques in this model to contain more collagen deposits, and this absence of increased collagen deposition in the larger plaques of ILC2KO mice coupled with less Arginase-1 expression might indicate disrupted tissue repair mechanisms".

3. Moreover, the authors now show a drop in ILC2 levels in the periphery, but no change in the local aortic populations. How, then, do the authors explain the drop in IL-13 and IL-5 levels in the aorta?

Response: There is NO drop of IL5 and IL13 in the aorta after HFD. Please, see Fig 2D.

4. It is unclear why the authors compare ILC2KO-IL-5 groups with ILC2WT groups, but separate ILC2KO-IL-13 mice into a separate figure. Is there a reason why ILC2KO-IL-13 mice were not directly compared with ILC2WT mice?

Response: Those were separate experiments. This is now clearly indicated in the manuscript on page 13: "Therefore, complementing the observations with ILC2 sourced IL-5, we examined the function of ILC2-derived IL-13 in a separate set of experiments by reconstituting BMT recipients with 80% ILC2^{KO} and either 20% IL-13⁺ or 20% IL-13^{KO}".

5. The authors show a significant drop in atherosclerosis in ILC2KO-IL-5+ mice, but IL-5 had little to no effect on macrophage phenotype. The authors do not provide a potential explanation as to why IL-5 can now rescue the ILC2KO phenotype.

Response: The ILC2KO-IL5+ mice are essentially like WT mice. Thus, they also produce IL13, and we suggest that they may alter macrophage phenotype at least through IL13 production. We acknowledged that we could not identify the mechanisms of IL5-dependent atheroprotection. Please, see page 14: "IL-5-dependent atheroprotection was limited to the thoracic aorta and could not be attributed to changes in macrophage phenotype or B1-dependent natural IgM production".

MINOR

In Supplemental Figure 4, there is no D to which the authors reference and there is no lesion size noted for B.

Response: corrected.

Reviewer #4 (replacement for original Referee #1)
Remarks to the Author:

The authors should adequately respect the previous papers (refs. 23, 26, and 27 of the revised manuscript) and reviewers' comments. Nevertheless, employment of Rora^{fl/fl};Il7ra^{Cre} mice is an important step to prove the importance of ILC2 in the disease progression, which added our understanding of the role of ILC2 in atherosclerosis. It is preferable for authors, however, to state that because ILC2 are tissue-resident cells, whole body irradiation may not deplete ILC2 present in peripheral tissues and it is difficult to completely deplete ILC2 in Rora^{fl/fl};Il7ra^{Cre} mice.

Response: In fact, we performed that experiment and we presented the data in the manuscript. We clearly indicated that only 5% of tissue-resident ILC2s were not replaced after lethal irradiation and bone marrow transplantation. Please, see page 10: "We found that donor ILC2^{Thy1.1} fully reconstituted the lymphatic compartments (0% Thy1.2⁺ ILC2) as well as the majority of the GWAT tissue resident ILC2 (5% Thy1.2⁺) (Figure 4A)".

I found several minor points that need to be clarified.

1) Page 6, lines 2 and 5; Fig. 1B should not be referred to because Figure 1B does not show the surface phenotypes.

Response: corrected.

2) Page 6, line8; Figure 1C does not contain data on spleen.

Response: corrected.

3) Figure S1C and S2A are not referred to in the text. Figure S2B and S2C are referred to before Figure S2D and S2E.

Response: corrected.

4) Page 8 and Figure 2; the authors showed the decreased expression of IL-5 and IL-13 by ILC2 in MLN and spleen but not by those in aorta. Although the authors studied the role of ILC2s in aorta is involved in the pathogenesis of atherosclerosis, why do the authors pick up the phenotype of ILC2s in MLN and spleen?

Response: This was required to address both systemic and local changes in ILC2.

Reviewer #5 (replacement for original referee #2)
Remarks to the Author:

This revised manuscript examines the key role for ILC2 in atherosclerosis. The data are reasonably convincing if you consider that what is shown is a global, possibly systemic, role of ILCs in the disease. Apparently, they do this by secreting IL5 and IL13. The claimed measurement of ILCs in the aortic adventitia (Fig. 2B) while never showing the gating or plots therein confuses the issue of location as to where the ILCs are working.

Response: We have clearly indicated in the manuscript that future studies will be required to address the distinct contributions of peripheral versus aortic ILC2s to lesion development. Please, see page 14: "Future studies should try to understand the differential impact of HFD on peripheral versus aortic ILC2, and define their distinct contributions to limiting vascular inflammation and atherosclerotic lesion development".

We have now presented an example of FACS gating for ILC2 in the aortic adventitia (Fig S1D).

1. In fact, I am disappointed by the gating of ILCs overall in the plots that are shown and the fact that the adventitial gating is not even shown might suggest that it is poor or unconvincing. This should be shown or removed from the manuscript, with the manuscript being adjusted to take a more global (systemic) view of their role.

Response: We have now presented an example of FACS gating for ILC2 in the aortic adventitia (Fig S1D).

2. Also, the M2 polarization aspect of the story is especially unconvincing. The antibody used to Arg1 is not well validated and in any case, a single marker (Arg1 and iNOS) have been used to mark M2 and M1 macrophages, respectively. I believe that this aspect of the manuscript should be removed. There is no direct evidence of the mechanistic link and the actual evidence for skewed polarization is weak.

Response: We talked about M1- or M2- *associated* genes or markers. Protein expression was backed up by gene expression and assessment of several M1 and M2

associated genes in the in vitro experiments (Supplemental Figure 5). We now make it clear that Arg1 and NOS2 are 'suggestive' of 'M2-like' or 'M1-like' macrophages. There is no specific transcription factor for M1 or M2 macrophages, and no specific way to selectively delete or manipulate M1 and M2 macrophages. Thus, it is not possible to provide the direct evidence requested by the Reviewer that M2 macrophages were responsible for the change in the plaque phenotype. However, we have now acknowledged this limitation and discussed the potential mechanistic links between ILC2-derived type 2 cytokines, macrophage phenotype and changes in plaque size and composition. Please, see page 14: "IL-5-dependent atheroprotection was limited to the thoracic aorta and could not be attributed to changes in macrophage phenotype or B1-dependent natural IgM production. IL-13-dependent atheroprotection was associated with important changes in collagen deposition and macrophage phenotype, suggestive of alternative activation. However, the direct links between changes of macrophage phenotype and atheroprotection were not addressed".

REVIEWERS' COMMENTS:

Referee #3 submitted comments only to the Editor.